# *BAMboozle* removes genetic variation from human sequence data for open data sharing

Christoph Ziegenhain[1] & Rickard Sandberg [1✉]

The risks associated with re-identification of human genetic data are severely limiting open data sharing in life sciences, even in studies where donor-related genetic variant information is not of primary interest. Here, we developed *BAMboozle*, a versatile tool to eliminate critical types of sensitive genetic information in human sequence data by reverting aligned reads to the genome reference sequence. Applying *BAMboozle* to functional genomics data, such as single-cell RNA-seq (scRNA-seq) and scATAC-seq datasets, confirmed the removal of donor-related single nucleotide polymorphisms (SNPs) and indels in a manner that did not disclose the altered positions. Importantly, *BAMboozle* only removes the genetic sequence variants of the sample (i.e., donor) while preserving other important aspects of the raw sequence data. For example, *BAMboozled* scRNA-seq data contained accurate cell-type associated gene expression signatures, splice kinetic information, and can be used for methods benchmarking. Altogether, *BAMboozle* efficiently removes genetic variation in aligned sequence data, which represents a step forward towards open data sharing in many areas of genomics where the genetic variant information is not of primary interest.

[1] Department of Cell and Molecular Biology, Karolinska Institute, Stockholm, Sweden. ✉email: rickard.sandberg@ki.se

Modern omics methods have transformed life science research and especially sequencing-based approaches have seen a rise in popularity. Sharing sequencing data has been a major driver of innovation and has increased the pace of development in life sciences[1,2]. Free access to raw sequencing data is essential for research transparency and for large-scale integration of published data. With the maturation of technologies like single-cell RNA sequencing (scRNA-seq)[3], many fields have progressed to broad applications in healthy and diseased primary human samples.

However, sharing sequencing data generated from human donors comes with ethical concerns relating to data privacy, since the genetic variation that defines each of us can be used and misused to uniquely identify human individuals. Pioneering analyses have demonstrated the ability to infer identities from variants such as single-nucleotide polymorphisms (SNPs)[4] and short-tandem repeats[5]. Sequencing data without donor information becomes re-identifiable given a large enough reference database[6], and even when some of the variants are masked but combined with other demographic information[7]. It is therefore paramount to protect the study individuals' identity and genetic information, as reflected in recent legislation[8]. Accordingly, human sequencing data are deposited in controlled-access repositories (e.g., dbGaP[9] or EGA[10]), whereas alternative strategies, such as blockchain encryption[11] or masking algorithms[12], have yet to reach widespread adoption[13]. Controlled-access repositories are important for protecting sensitive genomic data by allowing data sharing only with specific researchers who have been granted access. However, the heavily increased barriers in sharing sequencing data are severely limiting study transparency, reproducibility, innovation, and development, especially as we enter the age of personalized medicine.

Having access to the raw sequence data is important for most re-analyses of published studies. Meta-studies that associate specific human genetic variation to disease or traits critically rely on the controlled access to human sequence variant data. However, re-analyses of published scRNA-seq data also benefit from having the access to raw sequence data[14,15], although not necessarily needing genetic variant information. For example, the integration of data from many studies (and individuals) improves the derivation of cell-type and cell-state-specific gene expression signatures[16]. In fact, many areas of functional genomics would benefit from the ability to openly share raw sequence data that were processed to remove the genetic variant information, while preserving all other aspects of the aligned reads.

To address the current limitations to data sharing of human sequencing data, we developed *BAMboozle*, a versatile and efficient program that reverts aligned read sequences (in Binary Sequencing Alignment Map (BAM) format) to the reference genome to efficiently eliminate the genetic variant information in raw sequence data. We demonstrate on single-cell genomics data sets that *BAMboozle* accurately and efficiently removed genetic variant information in sequence data, without sacrificing utility for downstream analyses.

## Results

**Strategy for stripping human sequence data of genetic information.** To lower the barriers in sharing sequence data, we propose, like others recently[17], to remove information on genetic variation that could be used to infer the identity from aligned reads and compromises the privacy of the donor (Fig. 1a). Genetic variation, including SNPs, indels, and short-tandem repeats, is the main compromising information to remove in such data. Provided that count data does not carry sufficient information to allow the identification of individual patients, data

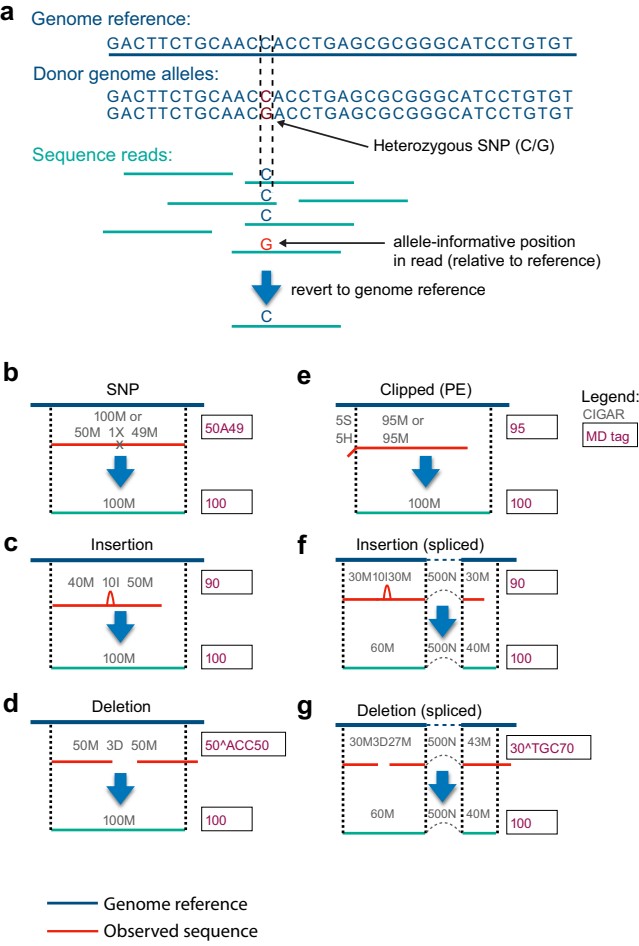

**Fig. 1 Schematic overview of the BAMboozle procedure. a** Sequenced reads typically harbor identifying genetic variants that are deviating from the reference genome. Data sanitation removes such variants by replacement with the reference sequence. **b** Schematic representation of an aligned 100 bp read (red) with an alternative allele at a single-nucleotide polymorphism (SNP) position is corrected to the reference base in output read (blue). Raw and corrected CIGAR (Compact Idiosyncratic Gapped Alignment Report) strings (with operator descriptions below) are shown over reads, and raw and corrected MD tags are shown to the right. **c** Aligned 100 bp read (red) with donor-specific insertion (relative to reference), with information as in **b**. **d** Aligned 100 bp read (red) with donor-specific deletion (relative to reference), with information as in **b**. **e** Aligned 100 bp read pair sequence (paired-end data) that was clipped at the 5′ end relative to reference. Note the correction in alignment end position as a result of the removal of clipped positions. **f** Spliced alignment that additionally contains an insertion not present in reference. The spliced alignment information is preserved during the sanitation of genetic variation. **g** Spliced alignment that additionally contains a deletion not present in reference. The spliced alignment information is preserved during the sanitation of genetic variation. CIGAR string operators: M, alignment match; I, insertion; D, deletion; N, skipped in reference; S, soft clipping; H, hard clipping; P, padding; =, sequence match; X, sequence mismatch.

processed to accurately remove all genetic variant information can—in principle—be shared openly according to the European General Data Protection Regulation (GDPR), fitting the criteria for "information which does not relate to an identified or identifiable natural person or to personal data rendered anonymous in such a manner that the data subject is not or no longer identifiable"[18]. If, however, it would turn out that count data does

carry considerable amounts of information about individuals, a procedure that eliminates all genetic variant information might only be useful for pseudonymization, which describes methods that reversibly reduce identifying information (GDPR recital 26)[19].

The removal of genetic variant information in typical sequence data needs to correctly handle multiple types of genetic variation (SNPs, insertions, deletions, short tandem repeats, immune haplotypes, etc.), present in different sequencing-based assays (e.g., RNA-seq, scRNA-seq, DNA-seq, ATAC-seq) from various sequencing platforms (Illumina, MGI, Pacific Biosciences, Oxford Nanopore Technologies) and strategies (e.g., single- or paired-end sequencing). At the same time, the remaining information such as alignment positions, custom Sequence Alignment Map (SAM) tags should be preserved as close to the original as possible, and the strategy should be agnostic to the sequence alignment tool used, as, for example, STAR[20] and BWA[21] report alignments differently that may affect the data processing. In contrast to a recent study[17], we reasoned it would be possible to write a single, efficient, versatile, and intuitive tool for the removal of genetic information from genomics data that can easily be integrated into existing infrastructures.

In brief, removal of genetic information from data can be implemented for sequenced and aligned reads stored in the ubiquitous BAM format by replacing donor genetic variation with the sequence of the reference genome (Fig. 1a). This procedure cannot be applied to the unmapped reads in the BAM file since donor-specific genetic information is harder to identify with no matching reference sequence. Unmapped reads are therefore discarded. In the simplest case, a fully aligned sequence read containing a SNP is replaced with the reference base (Fig. 1b). In the case of insertions in the donor sequence, the non-reference portion of the sequenced read is discarded and the remaining sequence is extended by the length equal to the insertion while keeping the 5' mapping position intact (Fig. 1c). Conversely, deletions are resolved by inserting the missing reference sequence and removing the equal numbers of bases from the 3' end of the altered sequence read (Fig. 1d). Portions of the read that may be soft- or hard-clipped are replaced by reference sequence, as the non-matching sequence could stem not only from technical artifacts such as adapters but also from genetic variation. In the case of reads starting with clipping, for single-end sequencing data the reference position of the read start is adjusted; however, this is not possible for paired-end reads because it would invalidate the mate-pair information (TLEN and PNEXT fields). Instead for paired-end reads, the clipped sequence portion is added to the end of the read (Fig. 1e). In RNA-seq data, spliced alignments are common and reads can span several exon–exon junctions. For these reads, both the 5' mapping position and the location of the splice junctions are preserved when scrubbing the sequence data of insertion and deletion events (Fig. 1f, g). In addition to the actual sequence field itself within the BAM file, information on the presence of genetic variation in the donor individual is stored in the CIGAR string and accessory fields. Therefore, the CIGAR field and MD tag are consequently corrected while removing each type of genetic variation (Fig. 1). Genetic information could be inferred from additional accessory fields, such as alignment score, mapping quality, and other alignment information (e.g., number of hits NH). To solve this, we update or remove these fields while leaving other auxiliary tags (e.g., sample information, cell barcode, gene identity, or custom flags) in place. While aiming to report data as close to the input sequence data as possible, resolving indel variation results in out-of-phase quality scores with respect to the base calls. The resulting privacy preserving BAM file is fully compliant with the SAM specifications[22] (as confirmed by *picard-tools* validation;

see "Methods") and thus smoothly compatible with existing bioinformatics tools and pipelines. We implemented our strategy for removing genetic variation in an open-source stand-alone Python script, called *BAMboozle*, that only requires the input BAM file and genome or transcriptome reference sequence used for the alignment.

Importantly, the removal of SNPs and indels with *BAMboozle* is not dependent upon user-supplied lists or databases of SNPs and indels, rather all single-nucleotide or indel differences relative to the reference sequence are replaced by the annotated reference bases. Thus, complete removal of SNPs and indels with *BAMboozle* is possible even though current databases of SNPs and indels are incomplete. This procedure effectively removes all types of genetic variant information, as sequenced reads aligning to the genome are all reverted to reference sequence, and the remaining sequenced reads that fail to align are discarded.

**Validating the absence of genetic information in BAMboozled sequence data.** To illustrate the effectiveness of *BAMboozle*, we analyzed a recently published 10× Genomics[23] dataset of scRNA-seq data generated from five equally abundant cell lines[24] derived from five different donors. After preprocessing the raw data with zUMIs[25], we summarized SNP coverage and assigned the 2937 high-quality (≥66% exonic, ≥25,000 reads) cells to their donor of origin using cellsnp-lite[26] and vireo[27]. Cellsnp-lite and vireo can identify sample-related genetic variant information with and without pre-defined list of SNPs or indels. Here these tools were used to quantify donor-related genetic variation present in the raw sequence data and after *BAMboozle* processing. We projected the scRNA-seq data in two dimensions using UMAP to visualize the overall structure of the cellular transcriptomic data (Fig. 2a). Clearly, cells from each cell line (and type) were distinctly grouped in the UMAP visualization indicating cell line-specific gene expression patterns, and importantly, each of the cells from each main cluster could be assigned to a single donor (colored in Fig. 2a) corresponding to the individual from which the respective cell line was derived. Next, we processed the BAM file with *BAMboozle* and repeated the donor assignment using cellsnp-lite and vireo and the same settings. Importantly, analysis of processed human sequence data failed to assign any cell to a donor (i.e., the data lacks a donor structure) (Fig. 2b). The transcriptome information, however, was completely intact enabling a meaningful analysis of the gene expression changes between the five studied cell lines. Finally, we investigated the number of reads in the data that had donor-specific information, as quantified by cellsnp-lite[26] and 23% of raw reads contained donor-specific information, whereas not a single read with donor-specific information remained after *BAMboozle* processing (Fig. 2c). Therefore, the complete lack of reads containing donor-related genetic variant information validated the removal of genetic information from human sequence data achieved with *BAMboozle*. As the cellsnp-lite analysis is currently not computationally feasible in unguided (whole-genome) mode for each individual cell, it was based on approximately 7.4 million most common human SNPs (see "Methods"). To confirm the absence of donor variation outside these previously known positions, we tabulated the coverage on every position of the genome over all cells using bcftools mpileup[28] and could confirm that the stored reads contained only reference sequence. We note that the results presented here present one of several possible state-of-the-art analyses and could be repeated with other workflows for SNP and indel detection.

Similar results were achieved when analyzing scRNA-seq data from human peripheral mononuclear cells (PBMCs) of four donors mixed together with a HEK293T cell line (derived from a separate donor) generated using the Smart-seq3 protocol[15] (Fig. 2d–f). To demonstrate that *BAMboozle* can also correctly remove genetic

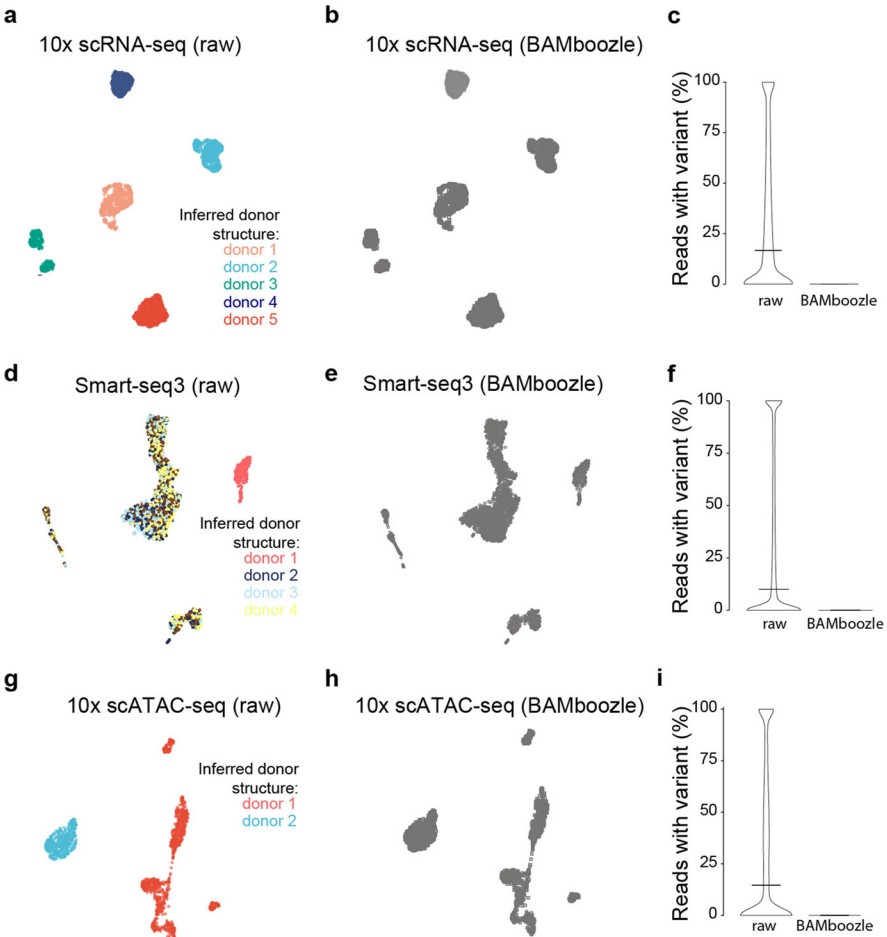

**Fig. 2 Processing of human scRNA-seq and scATAC-seq data with BAMboozle. a, b** Uniform Manifold Approximation and Projection (UMAP) plot of scRNA-seq data (10× Genomics) from five different cell lines (H2228, H1975, A549, H838 and HCC827; in total 2937 cells), colored according to donor identity inferred from the **a** raw and **b** processed sequencing data. **c** Percentage of 10× Genomics scRNA-seq reads containing alternate allele information at interrogated variant positions, for raw and processed data, respectively. All cells were combined and variants with sufficient coverage ($n = 75,461$) were used. **d, e** UMAP plot of scRNA-seq data (Smart-seq3) generated from PBMCs (4 donors) and HEK293T cells (in total 3129 cells), colored according to donor identity inferred from the **d** raw and **e** processed sequencing data. **f** Percentage of Smart-seq3 scRNA-seq reads containing alternate allele information at interrogated variant positions, for raw and processed data, respectively. All cells were combined and variants with sufficient coverage ($n = 1,503,063$) were used. **g, h** UMAP plot of scATAC-seq data (10× Genomics) generated from one PBMC donor and GM12787 cells (in total 1497 cells), colored according to donor identity inferred from the **d** raw and **e** processed sequencing data. **i** Percentage of 10× Genomics scATAC-seq reads containing alternate allele information at interrogated variant positions, for raw and processed data, respectively. All cells were combined and variants with sufficient coverage ($n = 95,127$) were used.

information from other data types, we next analyzed single-cell ATAC-seq (scATAC-seq) data that was derived from a single donor of PBMCs together with a lymphoblastoid cell line of a second donor. While the donor structure was clearly present in the raw BAM file (Fig. 2g), the *BAMboozle* procedure removed the ability to identify donor structures (using cellsnp-lite), while maintaining the cell-type separation (Fig. 2h). Reassuringly, processed sequence data did not contain a single read with genetic variation (Fig. 2i), again demonstrating the complete removal of SNPs and indels with *BAMboozle*.

**scRNA-seq data retain key utilities after removal of genetic information**. To demonstrate the utility of scRNA-seq data processed with *BAMboozle*, we applied several typical downstream analyses to the data. First, we demonstrate that the numbers of genes detected at different sequence read depths are not affected by the removal procedure (Fig. S1a), since this benchmarking is an important standard for method comparisons. Thereafter, we demonstrate that the cellular expression levels

were accurately computed, as exemplified by the two distinct clusters of B cells observed in the PBMC data (used above). The average gene expression levels (mean read counts) were virtually unchanged after removing genetic information (Fig. 3a, b). Unsurprisingly, testing for differential gene expression levels (see "Methods") yielded identical results (Fig. 3c, d), providing confidence that the *BAMboozle* procedure did not introduce unwanted noise. Next, we performed RNA velocity, an important analysis strategy that can capture and quantify the dynamic of cellular gene expression changes[29]. Of note, in cases where human scRNA-seq data are only available as count tables, RNA velocity analysis is not possible, as it requires the summarization of spliced, unspliced, and junction-spanning read counts from BAM files. We show that data processed with *BAMboozle* retains the same velocity information as the original data (Fig. 4), validating the scar-free removal of genetic information around and on splice junctions.

Naturally, analysis tools that are used for inferring clonal structures among cells are impossible to run on data lacking

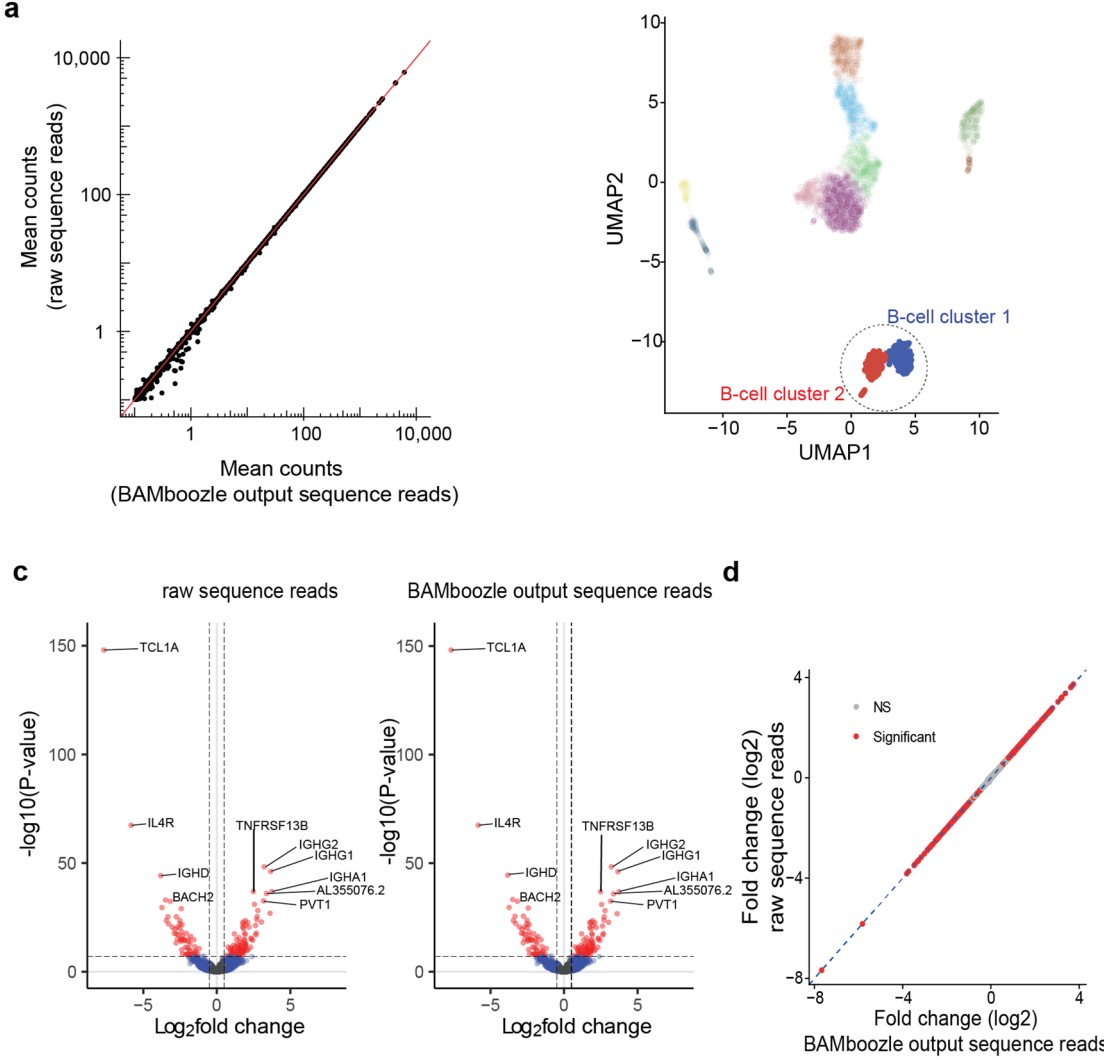

**Fig. 3 Accurate cellular transcriptomes in single-cell RNA-seq data after removal of genetic variants. a** Scatter plot showing accurate gene expression patterns in raw (*y*-axis) and *BAMboozle*-processed (*x*-axis) single-cell RNA-sequencing data. Data from 366 cells and 19,833 detected genes. **b** UMAP generated from processed sequence data maintains cell-type and state granularity, highlighting the two B cell clusters. **c** Volcano plots showing adjusted *p* values and log-fold changes (computed with limma-trend) in a comparison of B cell clusters 1 and 2, based on raw (left) and processed (right) sequence data, respectively. Genes are colored according to significance. **d** Scatter plots showing correlation in gene-level fold-changes estimated with limma-trend between B cell cluster 1 and 2.

genetic variation, since not a single informative read was present in the data after applying *BAMboozle*. Finally, we analyze the scRNA-seq dataset consisting of five distinct cell lines for the presence of copy number variations (CNVs). As the inference of CNVs using the *inferCNV* package (see "Methods") does not require additional sequence-derived information above the already widely shared gene expression count tables, we could infer CNVs even after applying *BAMboozle* (Fig. S1b).

**Comparison of *BAMboozle* and *ptools* performance**. Our implementation of *BAMboozle* differs in several key aspects from an alternative approach (called *ptools*) described by Gürsoy et al.[17]. First, *BAMboozle* does not restrict the removal of genetic information to the main chromosomes but rather takes all contigs in the user-provided genome reference. Alternative chromosome contigs and patches often contain highly polymorphic sites in the human genome, such as additional haplotypes for immunology-related genes such as major histocompatibility complex[30]. Second, whereas *ptools* retains original reads that are unmapped or

aligned on contigs without a reference sequence, we discard this information in *BAMboozle* to prevent leakage of clearly identifiable genetic sequence. Furthermore, we aim to minimize the information loss during the data processing by keeping splice sites and 5' mapping positions as intact as possible. We note that *ptools* does not retain information accurately in reads containing two or more spliced alignments (Fig. 5a), deletions in spliced alignments (Fig. 5b), and insertions in spliced alignments (Fig. 5c). The disruption of splice sites could potentially be used for identifying reads where the removal procedure was applied, and the amount of information leakage in such cases has not been systematically investigated. Furthermore, we aimed to provide an efficient and versatile implementation. To benchmark the strategies further, we processed a dataset consisting of 2.4 billion PE150 reads (the data used for Fig. 2d–f) with *ptools* and compared the performance with the processing with *BAMboozle*. The run-time of *BAMboozle* was 16.5-fold faster than *ptools* (5 h with *BAMboozle* vs. 80 h with *ptools*, Fig. 5d) while using 9.9-fold less storage space on disk at peak usage (Fig. 5e). Finally, since *ptools* did not remove genetic

**a**

Raw sequence data

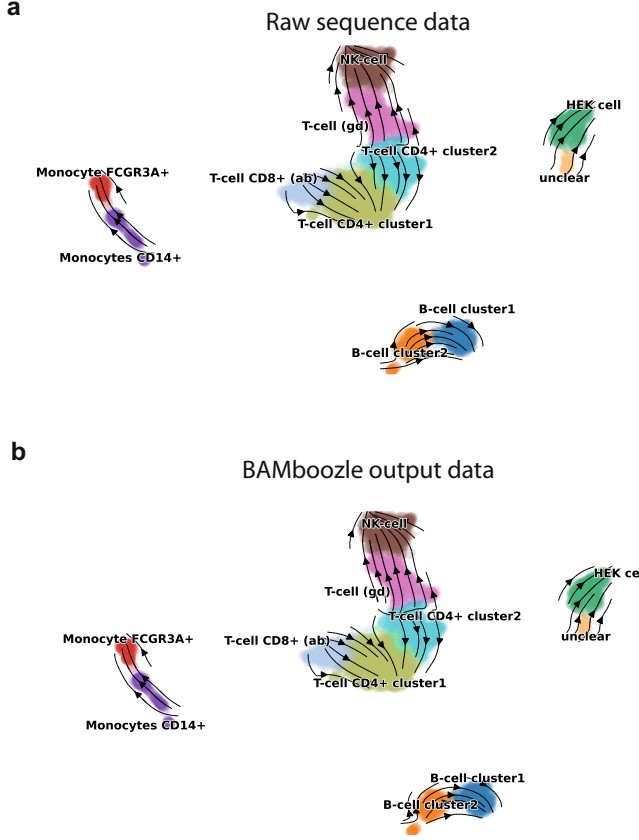

**b**

BAMboozle output data

**Fig. 4 Genotype-free RNA-seq data maintains splicing kinetics information. a** RNA velocity inferred from raw single-cell RNA-sequence data, with estimated cellular flows overlaid on top of the UMAP visualization (**b**). **b** RNA velocity analysis of single-cell RNA-sequencing data after removal of genetic variation with *BAMboozle* show similar cell-state flows as in the analysis run on raw sequence data (**a**).

variation present in alternative chromosome contigs, we note that 81,326 genomic positions containing donor-specific genetic variation remained after *ptools* processing (as compared to 0 such positions with *BAMboozle*).

## Discussion

The processing of personal data is protected through legislations, including the Health Insurance Portability and Accountability Act[31] in the USA and GDPR in Europe[18]. These apply to all aspects of processing of personal information, including the collection, processing, and storing of personal data of the individuals who donated biological samples, irrespectively of generating sensitive human sequencing data. In this study, we address the process of reverting human sequence data to pure reference genome sequence with the goal of open data sharing. All other aspects of data protection while processing personal information require careful separate attention[32]. Importantly, this type of anonymous data, i.e., data that lacks ability to trace the data to a natural person, is no longer under data protection legislation[18,19] and can in principle be openly shared. To this end, it is crucial that tools used for the processing of genomic information work accurately in order to truly remove sensitive donor-specifying genetic information before the community starts with open data sharing. Importantly, the procedure we developed in *BAMboozle* only removes genetic variant information. *BAMboozle* processed transcriptome data still contain CNV and gene expression levels, and their usefulness in identifying individuals,

particularly in combination with other data types, was not systematically investigated in this work. Nevertheless, *BAMboozled* transcriptome data contain similar information as the summarized data (e.g., sample- and gene-wise expression level count tables) that is currently openly shared in the research community. The widely shared summarized data at the read count level should be evaluated for their information content regarding donor identities, based on data alone and when taking donor demographic data into account.

Large-scale sequencing of human genomes has provided detailed information on the variation in the genetic sequences among individuals within and across populations. For example, the average human genome contains 4–5 million variant sites and 10–20,000 singletons (sites not found in other sequenced human genomes). Out of these variant sites, 99.9% are SNPs and short indels. The remaining variants are structural variations (approximately 2000 per genome) in the form of large deletions, CNVs, insertions of transposable elements, and nuclear mitochondrial variants[33]. Additionally, short-tandem repeats including minisatellite and microsatellite repeats can be used to identify individuals. With *BAMboozle*, we aim to remove genetic variation present in the sequence data by altering all variant sites to become the reference sequence that was defined by the researcher, typically standardized sequence assemblies such as GRCh38 curated by the Genome Reference Consortium[34]. The strategy we implemented correctly handles SNPs and indels and preserves the context in which these variants were identified (e.g., in spliced alignments). *BAMboozle* avoids making the altered sequences detectable by minimizing inconsistencies or correction "scars" in the processed BAM file, thus precluding a third person from identifying which sequences was corrected and the indirect inference of likely genetic variants. The *BAMboozle* strategy also handles more technical aspects of aligned sequencing data such as clipping and multimapped secondary alignments for spliced and regular alignments. Importantly, *BAMboozle* removes genetic variant information without relying on contemporary databases, and the strategy developed automatically removes all major kinds of genetic variation (SNPs, indels, repeat variants, microsatellite repeats, genomic rearrangements, translocations). As an example, insertions and deletions of all lengths are accurately eliminated since shorter insertions or deletions present within aligned reads are reverted to reference sequence (Fig. 1c), whereas reads with longer insertions or deletions that fail to align to the reference sequence are discarded.

Many areas of genomics rely on the genetic information obtained from each human donor, including genome resequencing and genome-wide association studies, and for these studies *BAMboozle* has no utility. However, the sharing of human sequence data reverted to reference genome sequence is highly relevant for many other areas of genomics, e.g., functional genomics. We envision that the exploration and enumeration of the cell types of human tissues would be greatly enhanced by *BAMboozle*, since it would facilitate integrative analysis of multiple datasets in manners compatible with batch corrections and integration strategies[35] to combat the effect of donor variation onto gene expression patterns. Benchmarking studies across scRNA-seq protocols, individuals, and conditions would also benefit from open data availability, for instance, by quantifying gene expression estimates from different sequence depths[14,36]. It is also worth noting that certain other aspects of functional genomics data are removed by this procedure, e.g., the ability to study allele-specific transcription including X-chromosome inactivation[37], reconstruct TCR and BCR repertoires, or the ability to infer clonal relationship between cells in cancer. Finally, we envision implementing an allow-list for one or a limited set of critical genomic positions in a dataset, so that these positions

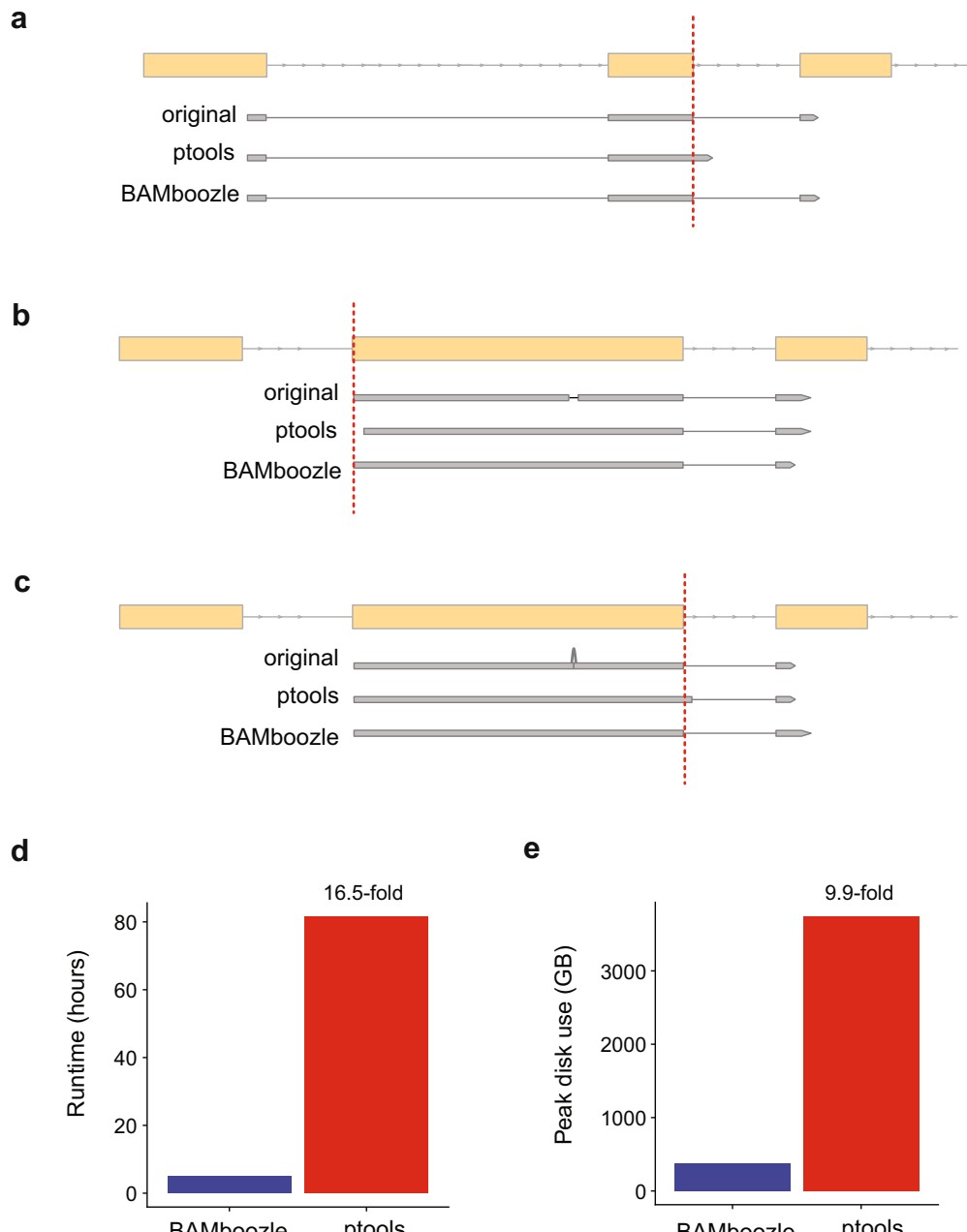

**Fig. 5 Comparison between *BAMboozle* and *ptools*. a–c** Spliced alignments with avoidable information loss during processing in *ptools*. Yellow boxes denote exons for gene models. Exemplary reads (original and processed by *ptools* or *BAMboozle*) are shown as gray boxes. Reference skip (*N*) for spliced alignments is denoted by gray lines. **a** Alignment covering several splice junctions is correctly reported by *BAMboozle* while *ptools* reports only the first splice site. **b** Spliced alignment with deletion. Information loss in *BAMboozle* is avoided by keeping 5′ mapping position and splice junction intact. **c** Insertion in a spliced alignment is resolved while keeping splice junction intact in *BAMboozle*. **d**, **e** Benchmarking of *ptools* and *BAMboozle* on 2.42 billion PE150 reads shows that run time for *BAMboozle* is **d** 16.5-fold faster (4 h 57 min vs 81 h 31 min) and **e** consumes 9.9-fold less storage space (378.5 GB peak vs 3741.1 GB peak).

have their genetic information intact, whereas all other genetic variations are removed. The number of variant positions to allow would require, however, in-depth analysis and calculation of privacy risks to ensure that the included genetic variation does not enable re-identification of the natural person.

The strategy we developed and implemented in *BAMboozle* is well aligned with the GDPR's evaluation of reasonable factors concerning the available and novel technology as well as the time and expertize required to attempt singling out individuals. However, the legal and ethical contexts of processing and sharing human sequence data are complex topics with differences across countries and legislations. Thus, researchers should make

informed decisions based on their local regulations and taking new developments in privacy research into account. Finally, the community would benefit from investigating the possibility of identifying sensitive, donor-identifying, genetic information from pure reference sequence count data. However, even if donor-specific information is detectable in gene expression data, it is currently not feasible to infer genetic information from expression data in humans, nor obvious that anything additional could be learned in addition to the data used for the search itself.

We here demonstrate how to achieve efficient and accurate removal of genetic information of human sequencing read data. The generation of such sequence data should re-enable open

sharing of sequence data in studies that do not require the genetic variation. This will, for example, be important for large publicly funded landmark projects, such as the Human Cell Atlas[38]. Ideally, controlled-access databases (e.g., dbGaP[9] or EGA[10]) could also directly offer free download of scrubbed data as a complement to the full genetic sequence data under controlled access. Finally, our versatile strategy for reliably reverting sequence data to the reference sequence is implemented into a user-friendly, efficient, and freely available tool present on GitHub (https://github.com/sandberg-lab/dataprivacy) and available from the PyPI repository (can be installed with: *pip install BAMboozle*). It is our hope that *BAMboozle* will facilitate data sharing of human sequence data for improved transparency and innovation in the life sciences.

## Methods

**Data sources**. Raw data in fastq format for the 10× Genomics v2 scRNA-seq experiment of five human cell lines (H2228, H1975, A549, H838, and HCC827) were obtained from the European Nucleotide Archive (accession: SRR8606521). Raw fastq data from scRNA-seq data generated using the Smart-seq3 protocol from PBMC and HEK cells were obtained from ArrayExpress (accession: E-MTAB-8735). scATAC-seq data for human PBMCs and GM12878 cells were downloaded as preprocessed (CellRangerATAC v 1.2.0).bam files from the 10× Genomics website (https://support.10xgenomics.com/single-cell-atac/datasets/1.2.0/atac_hgmm_1k_nextgem; https://support.10xgenomics.com/single-cell-atac/datasets/1.2.0/atac_pbmc_1k_nextgem) and concatenated for the remainder of the analysis.

**scRNA-seq data processing**. Raw fastq files were processed using zUMIs[25] (v 2.9.4 f), which relied on mapping of the reads using STAR[20] (v 2.5.4b) to the human genome (hg38) and quantification using Ensembl gene annotations (GRCh38.95). To generate UMAP plots, count tables were analyzed in Seurat[39] (v.3.1.5) following the standard workflow with default settings. In the case of Smart-seq3 data, the UMAP coordinates as previously published were used[15].

**BAMboozle workflow**. The data removal procedure involves modification of the observed read sequence to the reference genome sequence and the clearing of auxiliary tags.

Overview of considered cases and the associated strategy that is automatically being applied within the *BAMboozle* program:

1. SNPs: Mismatches to the reference (either explicitly *X* coded in the CIGAR value or within *M* matched segments) are replaced by the reference base.
2. Insertions (CIGAR *I*): The read sequence is extended by the length equal to the insertion, while keeping the 5' mapping position constant.
3. Deletions (CIGAR *D*): The missing reference sequence is inserted into the read, while removing an equal number of bases from the 3' end.
4. Clipping: Soft or hard clipped bases (CIGAR: *S/H*) are replaced by reference sequence of matching length. If reads start with clipped bases in single-end data, the reference position of the read start is adjusted, which is not possible for paired-end reads to conserve correct mate-pair information in the TLEN and PNEXT fields. Thus, for paired-end reads, the clipped sequence length is added at the end of the read.
5. Splicing: Splicing is observed and splice sites are conserved even in the case of deletions and insertions. In case of a deletion leading to a shift of length that is longer than the mapped sequence length in the last exon, this splice event is removed.
6. Multimapping: In default behavior, only primary alignments are emitted. The user can choose to keep secondary but we note that full protection from inference of genotypes cannot be guaranteed.
7. Unmapped reads: Unmapped reads cannot be cleared of genetic variation and are discarded in default settings.

As donor-related information could also be inferred from standard bam fields and auxiliary tags, the following changes are made:

1. CIGAR value is matched to the reference genome read sequence (Example: 100 M).
2. MD tags coding mismatches or deleted information are updated to the error-free sequence, if present (Example: 100).
3. NM and nM tags (edit distance to the reference) are cleared by replacement with 0.
4. Tags containing information on the alignment are discarded (MC, XN, XM, XO, XG).
   In --*strict* mode, the following tags are also changed:
5. Mapping quality is set to maximum/unavailable (255).
6. AS and MQ (alignment score/mapping quality) are set to read length.

7. NH (number of hits in the reference) is set to 1.
8. Discarding of the following tags: HI, IH, H1, H2, OA, OC, OP, OQ, SA, SM, XA, XS.

**Genotyping and donor inference**. Genotype-informative base coverage was summarized per cell over 7.4 million common variants (AF > 5% in the 1000 genomes project phase 3) with cellsnp-lite[26] (v 1.2.0) using --UMItag None --minCOUNT 10 settings. The resulting sparse VCF file was loaded into vireo[27] (v 0.4.2) and donor deconvolution performed using the appropriate --nDonor flag ($n = 2$ for scATAC, $n = 5$ for 10× scRNA-seq, and $n = 6$ for Smart-seq3).

**Validation BAM specification**. The output of *BAMboozle* from the Smart-seq3 scRNA-seq dataset was validated for compliance with the SAM specification[22] using the picard-tools (http://broadinstitute.github.io/picard) ValidateSamFile command with the following exceptions: --IGNORE MATE_NOT_FOUND, RECORD_MISSING_READ_GROUP, MISSING_READ_GROUP. No errors or warnings were observed.

**Comparison to *ptools***. The latest version of *ptools* available at the time of this manuscript (git commit # a684509 from 23 Nov 2020) was downloaded from GitHub https://github.com/ENCODE-DCC/ptools. We applied the "genome" workflow as all RNA-seq and ATAC-seq data were aligned to the human reference genome as described above. The following changes were applied to the *ptools* code in order to avoid errors during the processing:

1. **makepBAM_genome.sh:** lines 13–14 were changed to avoid requirement of global installation of ptools script in PATH:
   13: python3 $(dirname $(readlink -f $0))/getSeq_genome_wN.py "${reference_fasta}" header.txt withN.sam | samtools view -h -bS - > withN.p.bam
   14: python3 $(dirname $(readlink -f $0))/getSeq_genome_woN.py "${reference_fasta}" header.txt withoutN.sam | samtools view -h -bS - > withoutN.p.bam
2. **getSeq_genome_wN.py** and **getSeq_genome_woN.py:** lines 56/57, respectively, were changed to apply to Ensembl-formatted chromosome names used in this work (instead of UCSC nomenclature only). We note that the default ptools implementation would emit all data using Ensembl chromosome names unchanged.
   57: if chrom in ['1', '2', '3', '4', '5', '6', '7', '8', '9', '10', '11', '12', '13', '14', '15', '16', '17', '18', '19', '20', '21', '22', 'X', 'MT', 'Y']:
3. **getSeq_genome_wN.py:** line 78 moved before line 56, to avoid an error while writing the unchanged sequences that ptools does not discard.
   57: nColpbam = len(pbam)
4. **getSeq_genome_wN.py** and **getSeq_genome_woN.py:** lines 89–90/53–54, respectively: misformatted tab characters were corrected to avoid indentation errors.

Finally, *ptools* was run using the following command: "./ptools/genome/makepBAM_genome.sh Smartseq3.filtered.Aligned.GeneTagged.UBcorrected.sorted.bam /home/chrisz/resources/genomes/Human/hg38.primary_assembly.fa"

Checks for residual donor-related variation were performed using the bcftools (v1.7) mpileup command. The resulting compressed VCF files were filtered for the presence of alternate alleles with read support using bcftools view --min-alleles 3.

**Differential expression analysis**. Read count tables for introns and exons derived from the original data and *BAMboozle* output data were used, and we selected cells assigned to the two known B cell clusters. Genes expressed above an average of ≥0.1 counts across all cells were kept for differential expression. Next, count tables were normalized using scran[40] (v1.18.7) and differential expression was tested for using limma-trend[41] (v3.46.0). Raw *p* values were corrected using the Benjamini–Hochberg method.

**Inference of CNVs**. UMI count tables from the original and BAMboozle-processed data were used as input to the inferCNV package (v1.7.2; https://github.com/broadinstitute/inferCNV). CNVs were inferred using the recommended gene detection of 0.1 and using one arbitrarily chosen cell line cluster as the reference population, with all other settings kept at default values.

**RNA velocity**. Count matrices for spliced and unspliced reads were processed from original and *BAMboozle*-processed bam files using the velocyto python command line tool (v0.17.15)[29]. Recommended settings for Smart-seq data were used by running in velocyto run --without-umi --multimap mode. The resulting loom files were then further analyzed using the scVelo[42] implementation for RNA velocity (v.0.2.3), following the recommended default settings to perform gene selection, normalization, moment estimation, and estimation of RNA velocities. RNA velocities were overlaid onto UMAP representations using the stream embedding function.

**Reporting summary**. Further information on research design is available in the Nature Research Reporting Summary linked to this article.

## Data availability

The 10× Genomics v2 single-cell RNA sequencing experiment of five human cell lines was obtained from the European Nucleotide Archive (accession: SRR8606521). scRNA-seq data generated using the Smart-seq3 protocol were obtained from ArrayExpress (accession: E-MTAB-8735). Single-cell ATAC-seq data for human PBMCs and GM12878 cells was downloaded from the 10× Genomics website (https://support.10xgenomics.com/single-cell-atac/datasets/1.2.0/atac_hgmm_1k_nextgem; https://support.10xgenomics.com/single-cell-atac/datasets/1.2.0/atac_pbmc_1k_nextgem).

## Code availability

The python code has been deposited in GitHub (https://github.com/sandberg-lab/dataprivacy) and Zenodo[43] and can easily be installed through PyPI (https://pypi.org/project/BAMboozle).

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

## Acknowledgements

We are very grateful to Tim Dalessandri for coming up with the *BAMboozle* name and would like to thank the members of the Sandberg laboratory for fruitful discussions. This work was supported by grants to R.S. from the Swedish Research Council (2017-01062), the Knut and Alice Wallenberg Foundation (2017.0110), the Göran Gustafsson Foundation, and the Bert L. and N. Kuggie Vallee Foundation.

## Author contributions

Conceived the idea: R.S. and C.Z. Designed and developed *BAMboozle*: C.Z. Performed analyses and generated figures: C.Z. Wrote the manuscript: C.Z and R.S.

## Funding

## Competing interests

The authors declare no competing interests.
