## [Peer Review File · Nature Communications]

BAMboozle removes genetic variation from human sequence data for open data sharingReviewers' Comments:

Reviewer #1:

Remarks to the Author:

The authors address the need of sharing sequencing data which is often challenged by ethical concerns relating to the privacy of the donors. To tackle this challenge, the authors developed a tool for anonymizing human genetic data. It is based on the idea of replacing donor-related genetic information with according variants of the reference genome. To illustrate the effectiveness of the tool, a dataset containing single-cell RNA sequencing data was anonymized. The authors claim that the approach enables "open data sharing" of human sequence data.

Sharing genomic data is challenging and there is a great need for practical solutions. Therefore, the problem addressed by the authors is highly relevant. However, I have significant concerns whether the degree of protection provided by the method is sufficient, in particular for open data sharing, which requires truly anonymous data in a legal sense as no further technical or organizational safeguards can be implemented. Moreover, I'm also unsure about the practical relevance and novelty of the approach in general.

(1) The authors use the terms anonymization/anonymized and de-identification/de-identified interchangeably. However, these terms are often recognized as different concepts in the literature. To avoid misunderstandings, a consistent wording is critical. Moreover, wording depends on the legal basis referred to (e.g. the EU General Data Protection Regulation ("rendered anonymous") or the US Health Insurance Portability and Accountability Act ("de-identified")). It is also important to distinguish between "anonymisation", as the process of transforming data to reduce privacy risks and "anonymity" as the property of a dataset of containing non-personal data, the processing of which is typically not regulated. This paper could be helpful:

Chevrier R, Foufi V, Gaudet-Blavignac C, Robert A, Lovis C. Use and understanding of anonymization and de-identification in the biomedical literature: scoping review. *Journal of Medical Internet Research*. 2019;21(5):e13484.

(2) In addition to terminology, laws are also relevant to the claim made that the method is suitable for sharing data openly. At least a discussion of legal requirements in one or multiple jurisdictions and how those legal requirements relate to the anonymization process proposed is needed. If such a connection cannot be made, this must be described as a limitation of this work and it must be made clear that the question of whether data can be shared openly using the proposed approach remains open. The title should also be changed.

(3) Along the same line, the paper lacks a clear description from what types of threats the proposed approach is supposed to protect human sequence data. A wide range of privacy threats have been described in the literature and the following paper may serve as a starting point:

Mittos A, Malin B, De Cristofaro E. Systematizing genome privacy research: A privacy-enhancing technologies perspective. *Proceedings on Privacy Enhancing Technologies*. 2019 Jan 1;2019(1):87-107.

Most protection mechanisms work only within a specific context and the authors of this paper cite a paper by Shabani et al. addressing exactly this problem in a legal context and stating that assessing the identifiability of genomic data is highly dependent on context. More information on the threats considered and protection provided is needed.

(4) The proposed method aims to protect genetic data by modifying it. This will reduce the usefulness of the protected data. How exactly the approach aims to strike a good balance between protection and usefulness is not described. It might be the case that only DNA segments are modified that lie outside a scope of interest for a genotype under investigation, but this is not described in any detail. Even if this is true, the remaining sequence information may be identifiable and how to modify segments of

interest while maintaining anonymity would be the underlying problem that needs to be addressed to achieve the claimed goal of "open data sharing". Moreover, an analysis of the utility of output data should be added to the examples / evaluations presented. Without this additional information, the practicability and novelty of the approach proposed is hard to assess.

(5) I'm having problems interpreting Figure 2 (see also below). It looks like, as if the figure suggests that the protection provided by the technique is rather weak. Although the cell lines could not be assigned to their donor using cellSNP and Vireo, the clear emergence of clusters suggest that 1) a donor can easily be re-identified using external data, 2) the data is not protected from linkability (i.e. linkage of records belonging to one or multiple individuals).

In summary, the work is not convincing without (a) an description of relationships to legal requirements, (b) background information on the degrees of protection provided from which threats, (c) an analysis of utility of output data and the trade-offs taken.

Editorial remarks:

- Figure 1a: The figure is not easy to understand. What does it mean that three times cytosine becomes guanine?
- Figures 1 b-g: Tags are explained only in the caption, not in the text, which makes it a bit hard to follow the description.
- Figures 2b, e, h: Another type of plot may be more appropriate. Are the gray plots supposed to indicate that no inference is possible?
- Figures 2c, f, i: I'm not able to understand what these plots are supposed to show. Moreover, labels are missing on the x-axis.

Reviewer #2:

Remarks to the Author:

Ziegenhain and Sandberg have describe a tool which they call anonymiseBAM in light of the increase of human sequencing data generated and stored in access-controlled repositories with potential to share and release to researchers to augment into their studies.

The tool that they have created appears to be simpler than the one described Gürsoy et al. 2020 in terms of installation and accessibility. The justification for the motivation of such a tool is well made also in light of the Gürsoy paper with a desire to increase accessibility of human sequencing data while maintaining the privacy of individuals who donated their samples for sequencing.

Generalising the underlying sequence of the data (BAM files) back to the reference genome which was used to generate the BAM makes sense to achieve the motivations outlined and the authors have demonstrated this on three exemplar data sets in the single cell field through overlay of genotype information on standard UMAP of the original data and of the same data after processing with anonymiseBAM.

It is great that the author's show what the tool does to the BAM file to scrub the genetic information away including SNP's and Indels. Which are pertinent in RNA-Seq and ATAC-Seq.

The tool works and was easy to install with documentation at the cited websites.

I would like to see more description of the limitations of the anonymized data within the manuscript as there are some I can see that could catch end users of anonymizedBAM scrubbed files may have, wrongly expected how useful the data really is. By scrubbing the genetic sequence information all allelic information is lost which could be a useful insight for allele specific expression for example. That being said, all read counting quantification applications would work fine with anonymised BAM files,

which includes standard RNA-Seq and ATAC-Seq.

Reviewer #3:

Remarks to the Author:

With the widespread application of next generation sequencing and single-cell sequencing technologies, genetic and functional genomics data generation and analyses become common practice. Sharing of such data across the scientific community is important but encounters the barrier of privacy protection, as individuals' genetic variants are within or could be inferred from the data. Ziegenhain and Sandberg developed anonymizeBAM as a tool to sanitize sequencing reads from RNA-seq data by replacing the variants that could be used to infer individual genetic information. The study design is quite straightforward, and the content is easy to follow.

1. The major concern is that the novelty is limited due to the recent publication by Gursoy et al. Cell, 2020, where the proof of concept, theoretical issues and practical pipeline were clearly demonstrated in detail. The tool anonymizeBAM in this study might provide an alternative solution de-identifying sequencing data for general users, however, it is unclear how it distinguishes and outperforms the previous method by Gursoy et al.

2. Although it would be possible to develop a tool to anonymize all types of genomic data, the current tool in this study is more applicable to functional genomic data without necessarily to study genetic variants.

3. How does different type of variant inferred from genomic data contribute to the identification of individual privacy?

Reviewer #1 (Expertise: Data protection):

The authors address the need of sharing sequencing data which is often challenged by ethical concerns relating to the privacy of the donors. To tackle this challenge, the authors developed a tool for anonymizing human genetic data. It is based on the idea of replacing donor-related genetic information with according variants of the reference genome. To illustrate the effectiveness of the tool, a dataset containing single-cell RNA sequencing data was anonymized. The authors claim that the approach enables “open data sharing” of human sequence data.

Sharing genomic data is challenging and there is a great need for practical solutions. Therefore, the problem addressed by the authors is highly relevant. However, I have significant concerns whether the degree of protection provided by the method is sufficient, in particular for open data sharing, which requires truly anonymous data in a legal sense as no further technical or organizational safeguards can be implemented. Moreover, I’m also unsure about the practical relevance and novelty of the approach in general.

(1) The authors use the terms anonymization/anonymized and de-identification/de-identified interchangeably. However, these terms are often recognized as different concepts in the literature. To avoid misunderstandings, a consistent wording is critical. Moreover, wording depends on the legal basis referred to (e.g. the EU General Data Protection Regulation (“rendered anonymous”) or the US Health Insurance Portability and Accountability Act (“de-identified”). It is also important to distinguish between “anonymisation”, as the process of transforming data to reduce privacy risks and “anonymity” as the property of a dataset of containing non-personal data, the processing of which is typically not regulated. This paper could be helpful:
Chevrier R, Foufi V, Gaudet-Blavignac C, Robert A, Lovis C. Use and understanding of anonymization and de-identification in the biomedical literature: scoping review. *Journal of Medical Internet Research*. 2019;21(5):e13484.

We thank the reviewer for the helpful references and comments regarding definitions and terminology. Considering these comments, we revised the manuscript to:

1) remove the use of de-identification, as that has a rule-based definition in the HIPA Act and is mostly used in the context of personal information (not human sequence data).

2) we define our use of anonymous and anonymized in the revised manuscript. On Page 2 we write:

Anonymous data, “namely information which does not relate to an identified or identifiable natural person or to personal data rendered anonymous in such a manner that the data subject is not or no longer identifiable” and we refer to the GDPR.

Additionally, on page 3 we clarify how anonymized is used in this manuscript:

Thus, in this work we will use the term “anonymized” in the sense of GDPR recital 26 for data irreversibly rendered anonymous (as opposed to pseudonymization, which is a process to reversibly reduce identifying information), citing Recital 26 of GDPR.

With these improvements the revised manuscript is more stringent in the nomenclature and we clarify our intended use of the terms.

(2) In addition to terminology, laws are also relevant to the claim made that the method is suitable for sharing data openly. At least a discussion of legal requirements in one or multiple jurisdictions and how those legal requirements relate to the anonymization process proposed is needed. If such a connection cannot be made, this must be described as a limitation of this work and it must be made clear that the question of whether data can be shared openly using the proposed approach remains open. The title should also be changed.

The reviewer raises an important question regarding the legal context of using anonymizeBAM. To clarify this aspect, we added a new paragraph in the discussion on the use of anonymizeBAM in the legal contexts of GDPR. We chose to only relate anonymizeBAM to GDPR, as GDPR is the most stringent legislature and if anonymizeBAM would comply with GDPR, it would by default apply to other countries with less stringent legislature. The new paragraph was added to the discussion on page 6 (also pasted in below) and with this paragraph the legal context should be clear.

The processing of personal data is protected through legislations, including the Health Insurance Portability and Accountability Act²⁵ in the USA and General Data Protection Regulation (GDPR) in Europe¹³. These apply to all aspects of processing of personal information, including the collection, processing and storing of personal data on the individuals which donated biological samples, irrespectively of generating sensitive human sequencing data. In this study, we only address the process of generating anonymous human sequence data to enable open data sharing. All other aspects of data protection while processing personal information require careful separate attention²⁶. Importantly, the generation of anonymized data, i.e. data that has been irreversibly altered to remove the ability to trace the data to a natural person, is no longer under data protection legislation^{13,14} and can be openly shared. To this end, it is crucial that tools used for the anonymization of genomic information work accurately in order to truly remove all sensitive donor-specifying genomic information, before the community starts with open data sharing.

(3) Along the same line, the paper lacks a clear description from what types of threats the proposed approach is supposed to protect human sequence data. A wide range of privacy threats have been described in the literature and the following paper may serve as a starting point:

Mittos A, Malin B, De Cristofaro E. Systematizing genome privacy research: A privacy-enhancing technologies perspective. Proceedings on Privacy Enhancing Technologies. 2019 Jan 1;2019(1):87-107.

Most protection mechanisms work only within a specific context and the authors of this paper cite a paper by Shabani et al. addressing exactly this problem in a legal context and stating that assessing the identifiability of genomic data is highly dependent on context. More information on the threats considered and protection provided is needed.

We agree that a more explicit discussion of the types of sensitive genetic information in the human genome would strengthen our study and we therefore added the following new paragraph to the discussion (on page 6 and 7 of the revised manuscript, also pasted in

below). This information is important in order to fully understand the rationale of our anonymization strategy and we thank the reviewer for bringing up this question.

Large-scale sequencing of human genomes has provided detailed information on the variation in the genetic sequences among individuals of the same population and across populations. For example, the average human genome contains four to five million variant sites and 10-20,000 singletons (sites not found in other sequenced human genomes). Out of these variant sites, 99.9% are SNPs and short indels. The remaining variants are structural variations (approximately 2,000 per genome) in the form of large deletions (1,000), copy-number-variations (CNVs), insertions of transposable elements, and nuclear mitochondrial variants²⁷. Additionally, short tandem repeats (including mini and microsatellite repeats) can be used to identify individuals. With anonymizeBAM, we aim to remove all genetic variation present in sequence data by altering all variant sites to become the reference sequence that was defined by the researcher, typically the standardized sequence assemblies such as GRCh38 curated by the Genome Reference Consortium²⁸. The strategy we implemented correctly handles SNPs and indels, and preserves the context in which these variants were identified (e.g. in spliced alignments) while avoiding finding the altered sequences (i.e. no inconsistencies or correction “scars” are left on the processed BAM file) precluding a third person from identifying which sequences were corrected and the indirect inference of likely genetic variants. Moreover, our strategy handled more technical aspects of aligning sequencing data such as clipping and multimapped secondary alignments for spliced and regular alignments. We naturally did not attempt to remove copy-number variations (CNVs) in the anonymizeBAM tool, since this information is also present in the count files that are currently shared openly. To what extent gene expression data itself contains identifiable information is another open question in the field. Finally, the community would benefit from investigating the possibility of unbiased classifiers that attempt to identify sensitive, donor-identifying, genetic information from anonymized data. However, even if donor-specific information is present in gene expression data, it is questionable to what extent learning that information would enable a third party to learn anything important about the genotype of the investigated individual. Hence, we argue our approach is well-aligned with the GDPR’s evaluation of reasonable factors concerning the available and novel technology as well as the time and expertise required to attempt singling out individuals.

(4) The proposed method aims to protect genetic data by modifying it. This will reduce the usefulness of the protected data. How exactly the approach aims to strike a good balance between protection and usefulness is not described. It might be the case that only DNA segments are modified that lie outside a scope of interest for a genotype under investigation, but this is not described in any detail. Even if this is true, the remaining sequence information may be identifiable and how to modify segments of interest while maintaining anonymity would be the underlying problem that needs to be addressed to achieve the claimed goal of “open data sharing”. Moreover, an analysis of the utility of output data should be added to the examples / evaluations presented. Without this additional information, the practicability and novelty of the approach proposed is hard to assess.

This comment relates to the use of human sequence data rendered anonymous using anonymizeBAM. We think there are broad and urgent needs for this strategy, but there are of course trade-offs and certain analyses are no longer possible to conduct on the anonymous data. In the revised manuscript, we discuss this balance in a new paragraph on page 7 (also pasted in below).

The removal of donor-specific genetic information is clearly detrimental to certain applications, such as the study of genetic variation in relation to disease, for which there are no substitutes for obtaining the complete data from a controlled-access repository. The usability is however maintained for several other types of analyses. We envision that the exploration and enumeration of the cell types of human tissues would be greatly enhanced by anonymizeBAM, since the gene expression patterns per cell are preserved, as is the ability to use batch correction or integration strategies²⁹ to combat the effect of donor variation onto gene expression patterns. Benchmarking studies across scRNA-seq protocols, individuals and conditions, would also benefit from open data availability, for instance by quantifying gene expression estimates from different sequence depths^{30,31}. It is also worth noting however that other aspects of the functional genomics data are removed by this procedure, e.g. the ability to study allele-specific transcription including X-chromosome inactivation^{32,33}. Finally, we envision the implementing a whitelist for one or potentially multiple critical genomic positions in a dataset, so that the whitelisted positions have their genetic information intact whereas all other genetic variation is removed. The number of variant positions to whitelist requires however in-depth analysis to ensure that the included genetic variation does not enable re-identification of the natural person.

(5) I'm having problems interpreting Figure 2 (see also below). It looks like, as if the figure suggests that the protection provided by the technique is rather weak. Although the cell lines could not be assigned to their donor using cell SNP and Vireo, the clear emergence of clusters suggest that 1) a donor can easily be re-identified using external data, 2) the data is not protected from linkability (i.e. linkage of records belonging to one or multiple individuals).

We apologize for not explaining the Figure 2 properly. The data presented in Figure 2a is cellular transcriptomes from five different cell types that co-vary with them being derived from 5 different donors. We chose this data, as the proper donor identification should follow the cell line (and type), and because the donor structure we identify from the sequence data should map to the different cell clusters in the 2-dimensional projection.

The fact that the cells cluster into their proper cell line even after anonymizeBAM was applied reflects the fact that the transcriptome information (the expression patterns of all genes) was completely intact, enabling a meaningful analysis of the gene expression changes between the five studied cell lines, even after all donor-specific genetic variation was removed.

In the revised manuscript, we added more details into the results section that describe this experiment and results presented in Figure 2 (see page 4,5).

In summary, the work is not convincing without (a) an description of relationships to legal requirements, (b) background information on the degrees of protection provided from which threats, (c) an analysis of utility of output data and the trade-offs taken.

We thank the reviewer for bringing up these relevant comments, and we believe we have answered them with our revisions described above.

Editorial remarks:

- Figure 1a: The figure is not easy to understand. What does it mean that three times cytosine becomes guanine?

We have expanded on the Figure legends and text to better describe the intended representation. We also updated the Figure to add in the genotype of the individual (in this case a heterozygous SNP) of this hypothetical case to better clarify how DNA fragments with alternative bases were sequenced. The individual sequenced reads can then be derived from either of the alleles (thus having both cytosine or guanine at that position).

- Figures 1 b-g: Tags are explained only in the caption, not in the text, which makes it a bit hard to follow the description.

We worry that the results section will become harder to read if we describe all tags there also, and we prefer the original presentation of this information in the figure legend.

- Figures 2b, e, h: Another type of plot may be more appropriate. Are the gray plots supposed to indicate that no inference is possible?
Exactly, the cell-type information is unaffected by the anonymizeBAM processing (showing utility) whereas no donor-identifiable sequence fragment remained and therefore no possibility to infer a donor structure (or recover genetic sensitive information). We also clarified this detail in the results section describing Figure 2.

- Figures 2c, f, i: I'm not able to understand what these plots are supposed to show. Moreover, labels are missing on the x-axis.
We did have the x-axis labels "raw" and "anonymizeBAM" on the x-axis, although Figure 2 is fairly large and might not render properly in some PDF readers?

Reviewer #2 (Expertise: ATAC-Seq/RNASeq data analysis):

Ziegenhain and Sandberg have describe a tool which they call anonymiseBAM in light of the increase of human sequencing data generated and stored in access-controlled repositories with potential to share and release to researchers to augment into their studies.

The tool that they have created appears to be simpler than the one described Gürsoy et al. 2020 in terms of installation and accessibility. The justification for the motivation of such a tool is well made also in light of the Gürsoy paper with a desire to increase accessibility of human sequencing data while maintaining the privacy of individuals who donated their samples for sequencing.

Generalising the underlying sequence of the data (BAM files) back to the reference genome which was used to generate the BAM makes sense to achieve the motivations outlined and the authors have demonstrated this on three exemplar data sets in the single cell field through overlay of genotype information on standard UMAP of the original data and of the same data after processing with anonymiseBAM.

It is great that the author's show what the tool does to the BAM file to scrub the genetic information away including SNP's and Indels. Which are pertinent in RNA-Seq and ATAC-Seq.

The tool works and was easy to install with documentation at the cited websites.

I would like to see more description of the limitations of the anonymized data within the manuscript as there are some I can see that could catch end users of anonymizedBAM scrubbed files may have, wrongly expected how useful the data really is. By scrubbing the genetic sequence information all allelic information is lost which could be a useful insight for allele specific expression for example. That being said, all read counting quantification applications would work fine with anonymised BAM files, which includes standard RNA-Seq and ATAC-Seq.

The reviewer is raising an important issue, to what extent is the utility of the anonymizeBAM processed sequence data removed or intact. In the revised manuscript we added a discussion paragraph on this issue, that is also pasted below.

The removal of donor-specific genetic information is clearly detrimental to certain applications, such as the study of genetic variation in relation to disease, for which there are no substitutes for obtaining the complete data from a controlled-access repository. The usability is however maintained for several other types of analyses. We envision that the exploration and enumeration of the cell types of human tissues would be greatly enhanced by anonymizeBAM, since the gene expression patterns per cell are preserved, as is the ability to use batch correction or integration strategies²⁹ to combat the effect of donor variation onto gene expression patterns. Benchmarking studies across scRNA-seq protocols, individuals and conditions, would also benefit from open data availability, for instance by quantifying gene expression estimates from different sequence depths^{30,31}. It is also worth noting however that other aspects of the functional genomics data are removed by this procedure, e.g. the ability to study allele-specific transcription including X-chromosome inactivation^{32,33}. Finally, we envision the implementing a whitelist for one or potentially multiple critical

genomic positions in a dataset, so that the whitelisted positions have their genetic information intact whereas all other genetic variation is removed. The number of variant positions to whitelist requires however in-depth analysis to ensure that the included genetic variation does not enable re-identification of the natural person.

Reviewer #3 (Expertise: single-cell transcriptomics in humans):

With the widespread application of next generation sequencing and single-cell sequencing technologies, genetic and functional genomics data generation and analyses become common practice. Sharing of such data across the scientific community is important but encounters the barrier of privacy protection, as individuals' genetic variants are within or could be inferred from the data. Ziegenhain and Sandberg developed anonymizeBAM as a tool to sanitize sequencing reads from RNA-seq data by replacing the variants that could be used to infer individual genetic information. The study design is quite straightforward, and the content is easy to follow.

1. The major concern is that the novelty is limited due to the recent publication by Gursoy et al. Cell, 2020, where the proof of concept, theoretical issues and practical pipeline were clearly demonstrated in detail. The tool anonymizeBAM in this study might provide an alternative solution de-identifying sequencing data for general users, however, it is unclear how it distinguishes and outperforms the previous method by Gursoy et al.

We developed anonymizeBAM out of a need to share generated human sequence data, and apparently in parallel to the work of Gürsoy et al (and their implemented tool called: *ptools*). However, having read their paper we realized certain problems that urged us to finalize and submit the present work.

In response to this comment, we embarked to compare the strategies and benchmark the methods directly on a scRNA-seq dataset. The results are presented in new Supplementary Figure 1 of the revised manuscript (and also pasted in below).

We identify the following design problems:

1. *ptools* only considers main chromosomes (such as chr1, chr2, chrX, chrY) and outputs all reads coming from alternative chromosome contigs intact (there are plenty of these chromosomes). As a consequence, this resulted in over 81,000 genomic variants to be left un-anonymized in the *ptools* output as compared to anonymizeBAM that remove all genetic information. For users with chromosome names following Ensembl naming conventions (1, 2, ..., X, Y), *ptools* does not perform any anonymization without warning or error messages.
2. *ptools* leaves unmapped reads intact. This is another hazard, as the unmapped reads may contain sensitive genetic information and can often easily be re-mapped with less stringent alignment parameters.
3. *ptools* does not maintain the proper context (such as exon-intron junctions or starting 5' position intact) of the reads after correcting for indels. At least the following cases are incorrectly handled: i) reads containing two or more splice junctions, ii) deletion together with a splice junction, iii) insertion together with a splice junction. We graphically illustrate these problems in Supplementary Figure 1a (pasted in below).

Finally, processing a scRNA-seq sample with *ptools* and anonymizeBAM revealed that: i) *ptools* required 16-fold longer time to process this file (80 hours to finish, whereas anonymizeBAM required 4 hours; Supplementary Figure 1b)

ii) ptools required 10-fold more disk space (Supplementary Figure 1c)
 iii) importantly, ptools did not remove all sensitive information (Supplementary Figure 1d) as over 80,000 genetic variants were still present in their output.

iv) installation and running of ptools is not straightforward, as there are multiple conflicting versions on the developing lab's website and GitHub, as well as errors in the code that need to be dealt with prior to be able to perform anonymization.

We think the comparison above clearly illustrate the urgent need for a better anonymization tool for human genetic sequence data.

Supplementary Figure 1

Supplementary Figure 1. Comparison between anonymizeBAM and ptools.

(a-c) Spliced alignments with avoidable information loss during anonymization in ptools. Yellow boxes denote exons for gene models. Exemplary reads (original and anonymized by ptools or anonymizeBAM) are shown as grey boxes. Reference skip (N) for spliced alignments is denoted by grey lines. (a) Alignment covering several splice junctions is correctly reported by anonymizeBAM while ptools reports only the first splice site. (b) Spliced alignment with deletion. Information loss in anonymizeBAM is avoided by keeping 5' mapping position and splice junction intact. (c) Insertion in a spliced alignment is resolved while keeping splice junction intact in anonymizeBAM. (d-e) Benchmarking of ptools and anonymizeBAM on 2.42 billion PE150 reads shows that run time for anonymizeBAM is (d) 16.5-fold faster (4 hours 57

minutes vs 81 hours 31 minutes) and (e) consumes 9.9-fold less storage space (378.5 GB peak vs 3741.1 GB peak).

2. Although it would be possible to develop a tool to anonymize all types of genomic data, the current tool in this study is more applicable to functional genomic data without necessarily to study genetic variants.

The reviewer is correct. That is our intended use of anonymizeBAM. In the revised manuscript we added a new paragraph about the utility and trade-off with rendering anonymous data (also pasted in below).

The removal of donor-specific genetic information is clearly detrimental to certain applications, such as the study of genetic variation in relation to disease, for which there are no substitutes for obtaining the complete data from a controlled-access repository. The usability is however maintained for several other types of analyses. We envision that the exploration and enumeration of the cell types of human tissues would be greatly enhanced by anonymizeBAM, since the gene expression patterns per cell are preserved, as is the ability to use batch correction or integration strategies²⁹ to combat the effect of donor variation onto gene expression patterns. Benchmarking studies across scRNA-seq protocols, individuals and conditions, would also benefit from open data availability, for instance by quantifying gene expression estimates from different sequence depths^{30,31}. It is also worth noting however that other aspects of the functional genomics data are removed by this procedure, e.g. the ability to study allele-specific transcription including X-chromosome inactivation^{32,33}. Finally, we envision the implementing a whitelist for one or potentially multiple critical genomic positions in a dataset, so that the whitelisted positions have their genetic information intact whereas all other genetic variation is removed. The number of variant positions to whitelist requires however in-depth analysis to ensure that the included genetic variation does not enable re-identification of the natural person.

3. How does different type of variant inferred from genomic data contribute to the identification of individual privacy?

This is a highly relevant question. Most studies in the literature have explored the re-identification from genetic variations such as single-nucleotide polymorphisms (SNPs), insertion/deletions, short tandem repeats, immune-related haplotypes, and microsatellites. The re-identifiable information present in each of these types of genetic variation has not been thoroughly explored (at least not to our knowledge) and it might vary with the ethnicity of the sampled individuals for each study. In developing anonymizeBAM, we choose to remove all these types of genetic variation to enforce strict removal of sensitive information.

In the revised manuscript, we have added a new discussion paragraph on this subject:

Large-scale sequencing of human genomes has provided detailed information on the variation in the genetic sequences among individuals of the same population and across populations. For example, the average human genome contains four to five million variant sites and 10-20,000 singletons (sites not found in other sequenced human genomes). Out of these variant sites, 99.9% are SNPs and short indels. The remaining variants are structural variations (approximately 2,000 per genome) in the form of large deletions (1,000), copy-number-variations (CNVs), insertions of transposable elements, and nuclear mitochondrial

variants²⁷. Additionally, short tandem repeats (including mini and microsatellite repeats) can be used to identify individuals. With anonymizeBAM, we aim to remove all genetic variation present in sequence data by altering all variant sites to become the reference sequence that was defined by the researcher, typically the standardized sequence assemblies such as GRCh38 curated by the Genome Reference Consortium²⁸. The strategy we implemented correctly handles SNPs and indels, and preserves the context in which these variants were identified (e.g. in spliced alignments) while avoiding finding the altered sequences (i.e. no inconsistencies or correction “scars” are left on the processed BAM file) precluding a third person from identifying which sequences were corrected and the indirect inference of likely genetic variants. Moreover, our strategy handled more technical aspects of aligning sequencing data such as clipping and multimapped secondary alignments for spliced and regular alignments. We naturally did not attempt to remove copy-number variations (CNVs) in the anonymizeBAM tool, since this information is also present in the count files that are currently shared openly. To what extent gene expression data itself contains identifiable information is another open question in the field. Finally, the community would benefit from investigating the possibility of unbiased classifiers that attempt to identify sensitive, donor-identifying, genetic information from anonymized data. However, even if donor-specific information is present in gene expression data, it is questionable to what extent learning that information would enable a third party to learn anything important about the genotype of the investigated individual. Hence, we argue our approach is well-aligned with the GDPR’s evaluation of reasonable factors concerning the available and novel technology as well as the time and expertise required to attempt singling out individuals.

Reviewers' Comments:

Reviewer #1:

Remarks to the Author:

First, I would like to thank the authors for considering most comments I made in the first round! The paper has clearly improved regarding the issues I raised.

However, I still feel that the consideration of residual risks and the usefulness of output data of the method proposed is too superficial and that a formal analysis of both aspects (and the trade-offs taken) is lacking. Especially on the side of risks, I find this inappropriate, as the authors claim that output data fulfills anonymity in the sense of the GDPR and is suitable for publishing it openly. I am not convinced of this without a more in-depth analysis of residual risks. I also have doubts that a one-size-fits-all approach, as described in the paper, is adequate, even when only considering the privacy risk perspective. I would also like to point out that protection against "singling out" or protection from the possibility of others learning "useful information" about donors is not necessarily sufficient (cf. e.g. Opinion 05/2014 on Anonymisation Techniques by the Article 29 Data Protection Working Party [1]). Also the authors acknowledge that there is a range of open issues that have yet to be investigated, for example "To what extent gene expression data itself contains identifiable information is another open question in the field."

One easy way to resolve this would be to scale back the claims of the article a bit and to focus on the description of a "rules-of-thumb" method for reducing privacy risks. Open data sharing as well as anonymity in the legal sense could then be described as a potential future application that would probably need more research to be realized.

I would also like to point out that the paper by Gürsoy et al. is much more cautious with regard to the claims made ("Reduce Private Information Leakage"). This is despite the fact that a more systematic approach was taken with a formal and experimental analysis of risks and effects on data utility. While I must admit that I am lacking the biological and genetic background knowledge to fully understand the details of both approaches, this strengthens my belief that some paragraphs in this article (and especially the title) should be formulated with a little more caution.

[1] https://ec.europa.eu/justice/article-29/documentation/opinion-recommendation/files/2014/wp216_en.pdf

Reviewer #2:

Remarks to the Author:

Thank you for addressing my issue after the first version of the manuscript. I believe it has been improved to my satisfaction.

Reviewer #3:

Remarks to the Author:

The authors have addressed all the concerns.

Reviewer #1 (Remarks to the Author):

First, I would like to thank the authors for considering most comments I made in the first round! The paper has clearly improved regarding the issues I raised.

However, I still feel that the consideration of residual risks and the usefulness of output data of the method proposed is too superficial and that a formal analysis of both aspects (and the trade-offs taken) is lacking. Especially on the side of risks, I find this inappropriate, as the authors claim that output data fulfills anonymity in the sense of the GDPR and is suitable for publishing it openly. I am not convinced of this without a more in-depth analysis of residual risks. I also have doubts that a one-size-fits-all approach, as described in the paper, is adequate, even when only considering the privacy risk perspective. I would also like to point out that protection against "singling out" or protection from the possibility of others learning "useful information" about donors is not necessarily sufficient (cf. e.g. Opinion 05/2014 on Anonymisation Techniques by the Article 29 Data Protection Working Party [1]). Also the authors acknowledge that there is a range of open issues that have yet to be investigated, for example "To what extent gene expression data itself contains identifiable information is another open question in the field."

One easy way to resolve this would be to scale back the claims of the article a bit and to focus on the description of a "rules-of-thumb" method for reducing privacy risks. Open data sharing as well as anonymity in the legal sense could then be described as a potential future application that would probably need more research to be realized.

The reviewer raises a couple of questions that highlights residual risks with open data sharing and that further research is likely needed to investigate the legal and ethical consequences of open data sharing.

Re-reading the manuscript, we agree with the reviewer that certain formulations might have been too strong and we have tried to better balance the language at critical sentences throughout the manuscript, including the abstract, introduction, results and discussion. We are happy the reviewer brought up this additional point and we do think the study is now better described.

Regarding the additional risks with gene expression data, we clarified the discussion to make it clear that this data is currently openly shared without considerations for donor-sensitive information. Therefore, the anonymized sequence data we are producing would contain the same residual information (gene expression counts per cells or tissues) that are already being shared openly, thus not providing additional risks. We have also clarified this point in the discussion.

I would also like to point out that the paper by Gürsoy et al. is much more cautious with regard to the claims made ("Reduce Private Information Leakage"). This is despite the fact that a more systematic approach was taken with a formal and experimental analysis of risks and effects on data utility. While I must admit that I am lacking the biological and genetic background knowledge to fully understand the details of both approaches, this strengthens my belief that some paragraphs in this article (and especially the title) should be formulated with a little more caution.

Although Gürsoy used a more formal approach to quantifying privacy leakage, we note that their tool does not remove important genetic sensitive information, and in cases where information is removed, there are traces left that pinpoints locations of these corrections, which their analysis failed to identify.

As detailed in the answer above, we do agree with the reviewer that more cautious formulations is warranted, and we have revised the manuscript accordingly.

Reviewers' Comments:

Reviewer #1:

Remarks to the Author:

Thanks for addressing my comments!

Reviewer #4:

Remarks to the Author:

The work is certainly on an important and challenging problem. However, I see several major issues in this paper as I list below:

- First of all, unfortunately, there is no such a thing as "systematic anonymization" of genomic data (actually, we should not use that term for any type of data, since de-anonymization is always possible depending on an attacker's auxiliary knowledge).

- Information in the genome that identifies a donor is not something like a known region. The whole genome can be used to identify an individual depending on what the attacker knows about the individual. I am not sure which SNPs the authors are planning to mask. If they mask all SNPs, then the utility in the data will be completely lost (please see my later comment on this). If they hide some selective SNPs, there are lots of work on the literature showing the privacy risk under such selective hiding (e.g., inferring the masked SNPs from unmasked ones using the correlations between them).

- Privacy is validated/quantified using a bioinformatics tool only, not via established methods. There are several inference attacks on genomic data that can be used for de-anonymization. I would expect the authors at least run the obfuscated genome against such attacks to prove their statement.

- There is the well-known differential privacy concept to obfuscate data (there are several works using this for genomes as well), which provides statistical guarantees (also against de-anonymization). But, there is no mention of this in the entire paper.

- Paper needs a detailed utility analysis. Authors claim that there are still analyses that can be done on obfuscated data. If all SNPs will be masked, I do not see much utility in the remaining data. So, I think the paper certainly requires a detailed utility analysis.

- The works authors compare their proposed scheme with are not actually the seminal papers in this domain. I suggest the authors do a more depth literature review on this area. For example, hiding SNP information from short reads (BAM) files have been already explored. Please see the following work: https://link.springer.com/chapter/10.1007/978-3-642-54568-9_9. Also, authors should compare and position their work with existing attacks against genomic privacy (especially inference and de-anonymization attacks) and differentially-private approaches.

Reviewer #4 (Expertise: Privacy enhancing technologies, genomic data anonymization):

The work is certainly on an important and challenging problem. However, I see several major issues in this paper as I list below:

1. First of all, unfortunately, there is no such a thing as "systematic anonymization" of genomic data (actually, we should not use that term for any type of data, since de-anonymization is always possible depending on an attacker's auxiliary knowledge).

As we point out in the manuscript, we here solve the problem of removing genetic variation that makes up the inter-individual differences, namely SNPs, indels, microsatellite repeats etc, in an efficient and accurate manner. The discussion highlights what types of genetic information is anonymized and the few specific types that are not anonymized (e.g., CNVs), and similarly, what type of genetic information is currently openly shared (e.g., CNVs).

With *anonymizeBAM*, we do remove essentially all genetic variation (detailed in answers below), and the privacy concern with respect to the anonymized sequence data should be alleviated. We agree with the reviewer however that other aspects of preserving important meta-information of donors needs proper attention, and this we highlighted in the first discussion paragraph in our manuscript (page 8 of the revised manuscript).

2. Information in the genome that identifies a donor is not something like a known region. The whole genome can be used to identify an individual depending on what the attacker knows about the individual. I am not sure which SNPs the authors are planning to mask. If they mask all SNPs, then the utility in the data will be completely lost (please see my later comment on this). If they hide some selective SNPs, there are lots of work on the literature showing the privacy risk under such selective hiding (e.g., inferring the masked SNPs from unmasked ones using the correlations between them).

The reviewer has misunderstood our purpose with *anonymizeBAM* and the need for this tool in the research community. There are many large-scale human sequencing projects where sequence data needs to be shared although the SNPs are not primarily important (for example in the Human Cell Atlas and Lifetime projects, to just name two of the largest). *AnonymizeBAM* is the first tool that can efficiently and without leaving scars remove the sensitive information so that data generated from human tissues in these large consortia can be openly shared.

For this use, we indeed want to remove **all** SNPs and indels (and e.g. microsatellite repeat lengths), and the remaining data, often produced from functional genomics assays (e.g., RNA-seq, ATAC-seq, etc.), is highly useful (see details below). Furthermore, the removal of for example all SNPs and indels is possible without using a user-defined list of SNPs or indels (as by definition would be incomplete and therefore not provide full protection). Instead, *anonymizeBAM* removes all single nucleotide (and shorter insertion/deletion) that are detected as a difference between sequenced reads and the reference sequence (often the reference genome). Therefore, we can remove all such genetic variation without using a database, e.g., dbSNP. In the revised manuscript we expanded the results and discussion section to clarify this important point. The new discussion text reads:

“Importantly, anonymizeBAM does not rely on contemporary databases of SNPs and indels, rather we anonymize all such variation that differ with regards to the reference sequence. Therefore,

complete removal of SNPs and indels is possible despite incomplete databases. Moreover, the anonymization strategy implemented in anonymizeBAM automatically removes other kinds of genetic variation. In fact, insertions and deletions of all sizes are corrected for, since shorter insertions or deletions present within an aligned read is corrected for (see Figure 1c), whereas reads with longer insertions or deletions are discarded since they fail to align to the reference sequence. Therefore, the anonymizeBAM strategy effectively removes genetic variation present in microsatellite repeat lengths (and internal polymorphisms) without the need for a user-defined list of satellite positions or meta information.”

The fully anonymized data is intended to be openly shared (as the sensitive information has been removed) and used for the following important analysis questions (just to name a few areas of importance):

- 1) Study cell-type, sub-type and cell-state definitions and associated gene expression patterns from large-scale single-cell RNA-seq and ATAC-seq studies.
- 2) Perform meta-analysis across many data sets to improve point 1 above
- 3) Learn about alternative splicing or alternative use of promoters and polyadenylation sites for each cell, cell-type etc using population-level or single-cell RNA-sequencing.
- 4) Potential to use RNA velocity to study the directionality of cell-states by quantifying intronic and exonic reads for each gene and cell (very important for single-cell RNA-sequencing)
- 5) Potential to link RNA, ATAC-seq, splicing data for integrated analysis of the genetic programs inside each cell type etc.
- 6) Benchmarking studies on how different single-cell genomics methods perform in relation to other methods.

By these examples, we wish to highlight that the reviewer’s claim “If they mask all SNPs, then the utility in the data will be completely lost (please see my later comment on this)” is unfortunately very incorrect.

We agree that our tool is not useful in all areas of genomics, for example in areas where genetic variation is directly analyzed against traits or disease the raw data with highly protected access is required (and other strategies to obfuscate data become relevant). This was mentioned in the previous version of the manuscript and have in the revised manuscript been clarified more extensively.

Altogether, in the revised manuscript we have made an effort to clarify the utility of anonymized human sequence data throughout the text and in the new analysis that showcase important examples of clear utility (see details in the answer to comment 5 below).

3. Privacy is validated/quantified using a bioinformatics tool only, not via established methods. There are several inference attacks on genomic data that can be used for de-anonymization. I would expect the authors at least run the obfuscated genome against such attacks to prove their statement.

We removed **all** SNPs and indels as detailed above. To verify that we indeed successfully removed the SNPs and indels, we used bcftools, cell-snp-lite and vireo, which are widely used tools for finding donor-related genetic variation in large-scale sequence data. Importantly, these tools can be run with and without user-defined lists of SNPs and indels, and we ran them without user-defined lists

(where possible) for them to capture and quantify any deviation from reference. Indeed, using these tools we observed not a single read with genetic information in the anonymizeBAM processed files, which demonstrated the accuracy of anonymizeBAM. As all reads with SNPs and indels were removed (as well as all other genetic variation that deviate from reference sequence), there is little remaining genetic information to attempt an inference attack on, and more importantly nothing to be learned from an inference attack.

We have in the revised manuscript clarified that bcftools, cellsnp-lite and vireo indeed can be used without user-defined lists of SNPs or databases, and that this analysis validated the accurate removal of all SNPs and indels present in our human sequence data sets.

After the *anonymizeBAM* processing there is therefore no information left to obfuscate.

4. There is the well-known differential privacy concept to obfuscate data (there are several works using this for genomes as well), which provides statistical guarantees (also against de-anonymization). But, there is no mention of this in the entire paper.

Again, the whole point of *anonymizeBAM* is to remove **all** genetic variation, including SNPs, indels and microsatellite repeats, rather than masking a subset or using a strategy to obfuscate data. After *anonymizeBAM* processing the data that contain only reference sequence information can likely be openly shared.

5. Paper needs a detailed utility analysis. Authors claim that there are still analyses that can be done on obfuscated data. If all SNPs will be masked, I do not see much utility in the remaining data. So, I think the paper certainly requires a detailed utility analysis.

Please see our answer to comment 2 above. We remove all genetic variation relative to reference (including SNPs, indels, microsatellite repeats etc) and the anonymized data is still highly useful in functional genomics (although obviously not in e.g. population genetics or genome resequencing studies). In the provided manuscript discussion paragraph, we highlighted the areas for which *anonymizeBAM* is useful and where it will not be of use.

In the revised manuscript, we have added several additional examples of the utility of human sequence data processed with *anonymizeBAM*. We specifically show that:

1) RNA velocity (tool from here: <http://velocyto.org/>) analysis was used to quantify fully spliced and unspliced reads per gene and cell, in order to infer splicing kinetics per gene, which is used to estimate cell directionality. The anonymized files had similar information as the raw data, validating the correct resolving of genetic variation in and around splice sites with *anonymizeBAM*.

2) We show that the anonymized BAM files can be used for saturation analysis, e.g. to report number of genes or molecules per sequence depth, important for method benchmarking studies.

3) We show that CNVs can be inferred both on the full input data and after processing with *anonymizeBAM*.

4) We show that differential expression analysis is possible on anonymized data. To this end we exemplify the top differentially expressed genes between two clusters of B cells are identified with the same power in anonymous data as with the full data set, confirming that *anonymizeBAM* does not distort the mapping positions of reads.

We also tried running the anonymized BAM files on analysis tools that should not longer be feasible after removing genetic variation, including the Cardelino (McCarthy et al., 2020) and HoneyBADGER (Fan et al., 2018), tools used for inferring clonal structures among cells. However, since not a single informative read was present these tools naturally crash on anonymized sequence data, which demonstrates the point that analyses that rely on allelic differences or other genetic variation is not longer feasible after anonymization.

6. The works authors compare their proposed scheme with are not actually the seminal papers in this domain. I suggest the authors do a more depth literature review on this area. For example, hiding SNP information from short reads (BAM) files have been already explored. Please see the following work: https://link.springer.com/chapter/10.1007/978-3-642-54568-9_9. Also, authors should compare and position their work with existing attacks against genomic privacy (especially inference and de-anonymization attacks) and differentially-private approaches.

We thank the reviewer for mentioning the above study that we were not aware of. This citation has been added to the revised manuscript.

We note however, that the above-mentioned study focuses on an encrypted database to which queries can be made and masked sequence data is returned. This is very much in line with several ongoing efforts, although completely orthogonal to *anonymizeBAM* where the goal is to share data directly and openly after removing all SNPs, indels and e.g. microsatellite repeats. With this strategy, the remaining sequence data contains so little genetic variation (essentially nothing) that attacks trying to connect it to a person are futile and can also not lead to additional sensitive information (eg. disease-relevant SNPs) being learned anyways.

Editorial questions

(1) Even more careful wording when reporting the anonymity of the data outputted by `anonymizeBAM`, as none of the analyses presented exclude the possibility of combining the remaining genomic information with other data types to infer subject identity. This point was previously brought up by R1 in some form. While the reviewer suggests you actually explore how secure the data is by combining it with other data types, we appreciate this might be difficult if you do not have the right data for that. At least the cell lines you use in the current examples might not have been reported with much other subject-specific data, in which case, it's understandable that such an analysis is -in fact- not possible.

We have clarified the extent of anonymization obtained with `anonymizeBAM` in a new discussion paragraph, that should alleviate this concern (pasted in below):

“Importantly, the removal of SNPs and indels with `anonymizeBAM` is not dependent upon user-supplied lists of SNPs or indels, rather all single-nucleotide or indel differences relative to the used reference sequence is replaced by reference sequence. Thus, complete removal of all SNPs and indels with `anonymizeBAM` is possible even though current databases of SNPs and Indels are incomplete. Moreover, the anonymization strategy implemented in `anonymizeBAM` automatically removes other kinds of genetic variation. Insertions of all sizes are corrected for, since shorter insertions present within an aligned read is corrected for (Figure 1c) whereas longer insertions that results in the inability to align the sequence at all are discarded. Likewise, deletions of all lengths is removed if captured within an aligned read. Therefore, the `anonymizeBAM` strategy effectively removes genetic variation present in microsatellite repeat lengths (and internal polymorphisms) without the need for a user-defined list of satellite positions or meta information.”

As for possibly linking the anonymized sequences to donor-related information, there would be nothing to gain from it, as the anonymized sequences are stripped of essential genetic variation and are indistinguishable from the human reference sequence.

(2) An explanation of what SNPs and other genomic features were replaced, and why they were chosen.

We remove **all** genetic variation detected in the sequences compared to a reference sequence, that includes SNPs, insertions and deletions, microsatellite repeats and more. Importantly, this is achieved in a manner not relying on enumerated cases of known genetic variation in contemporary databases (such as SNPdb). In the revised manuscript we have clarified this in the discussion (as was pasted in the answer above),

(3) A utility analysis, which we believe is well within your laboratory's expertise, and is something we hope to be extended through a few examples. These examples need to demonstrate situations in which the "anonymized" data does not affect downstream analyses typically done with (sc-)RNASeq data, and one or more situation where it does (for example, exploring the clonality of patient cancers using sc-RNASeq data).

We have made a serious effort to expand on the utility of anonymized human sequence data in the revised manuscript. To this end we have added various exemplary analysis types that are supported by data processed with *anonymizeBAM*.

1) RNA velocity (tool from here: <http://velocyto.org/>) analysis was used to quantify fully spliced and unspliced reads per gene and cell, in order to infer splicing kinetics per gene, which is used to estimate cell directionality. The anonymized files had similar information as the raw data, validating the correct resolving of genetic variation in and around splice sites with *anonymizeBAM*.

2) We show that the anonymized BAM files can be used for saturation analysis, e.g. to report number of genes or molecules per sequence depth, important for method benchmarking studies.

3) We show that CNVs can be inferred both on the full input data and after processing with *anonymizeBAM*.

4) We show that differential expression analysis is possible on anonymized data. To this end we exemplify the top differentially expressed genes between two clusters of B cells are identified with the same power in anonymous data as with the full data set, confirming that *anonymizeBAM* does not distort the mapping positions of reads.

We also tried running the anonymized BAM files on analysis tools that should not longer be feasible after removing genetic variation, including the Cardelino (McCarthy et al., 2020) and HoneyBADGER (Fan et al., 2018), tools used for inferring clonal structures among cells. However, since not a single informative read was present these tools naturally crash on anonymized sequence data, which demonstrates the point that analyses that rely on allelic differences or other genetic variation is no longer feasible after anonymization.

Finally, we have made an effort to clarify in abstract and introduction the intended use of *anonymizeBAM* and have explicitly listed areas of genomics that would (or would not) benefit from using this anonymization tool. Therefore, we believe the utility of *anonymizeBAM* has been significantly clarified in the revised manuscript.

Reviewers' Comments:

Reviewer #2:

Remarks to the Author:

The Author's had addressed my initial concerns and upon reading the response to another additional reviewer on the manuscript, I think the responses there haven't been addressed.

It is nice to see the Author's articulate the utility of their tool in relation to applications that could be used on anonymised data even further, at the request of the additional reviewer, and to also demonstrate/or at least describe the demonstration of a tool that requires variant data, fail on the anonymised data.

Reviewer #3:

Remarks to the Author:

In this revision, the authors have acknowledged that the anonymizeBAM algorithm could be a straightforward solution to remove genetic information identifying individual for genomic applications without requirement of genetic variant information. The authors extended the algorithm to some single-cell sequencing applications and demonstrated its feasibility to keep the expression matrix and meanwhile anonymize the genetic identities. It is noteworthy that TCR/BCR information (sequence, diversity, and profiles) would be informative data to identify an individual. The reviewer would like to recommend the authors to point out that the algorithm is not applicable in the scenario where TCR/BCR information is remained for data analysis, especially in the sharing of 5' RNA-seq data.

Reviewer #4:

Remarks to the Author:

I am sorry again for my negative attitude, and please know that I am generally supportive and generous when reviewing papers. However, based on my privacy knowledge, privacy-related claims in this paper are wrong. This work can be published in a workshop to facilitate discussion maybe, but I think it would be a bad idea to publish this work in Nature Communications.

I read the new version of the paper and the rebuttal. I think it would be clearer if I respond to your comments in the rebuttal (which includes the changes in the paper anyway):

1) Alleviating the privacy concern is not a "systematic anonymization". Regardless of the metadata of the shared genome, what you share (even without the SNPs) is potentially vulnerable. The fact that most current attacks against genomic privacy today consider SNPs does not mean that by removing SNP data, you can provide anonymization to a shared genome. As a matter of fact, you claim that what is shared (after masking SNPs) is still useful. If the information is useful, this means that such information can be used to infer sensitive data about an individual. This is a common knowledge in privacy. That is why I suggested differential privacy in my previous review, which provides privacy guarantees against the attacks/vulnerabilities that are not even known today.

2) I don't think I misunderstood this. I completely agree that non-SNP data can be used for different analyses. However, to claim that "AnonymizeBAM is the first tool that can efficiently and without leaving scars remove the sensitive information" you need a formal privacy analysis, which you refuse conducting. Using the output of a bioinformatics tool to claim privacy guarantees is not an acceptable methodology in privacy domain. Your idea may be interesting; I do not want to discourage you in this direction. I suggest you try this publication in a more exploratory domain to discuss it with other researchers, before targeting Nature.

I have no problem how you remove the SNP, which I believe you do it in a proper way. My issue is: you claim removing SNPs will be enough to anonymize and safely share genomic sequences. This is also an ok claim as long as you justify it with a convincing privacy analysis. Unfortunately, such an analysis still does not exist in the paper.

If you claim that (which I believe) data after removing SNP is useful, you should (i) make a quantitative utility analysis, and most importantly (ii) conduct a privacy analysis for the remaining data to show that it is indeed safe to share. I sincerely wish your privacy analysis would give positive

results so we can safely start sharing genomes across institutions using your technique. However, without such an analysis it is not possible to acknowledge this claim. You can see the following works that does similar privacy analyses against some known inference attacks:

<https://dl.acm.org/doi/10.1145/2508859.2516707>

<https://pubmed.ncbi.nlm.nih.gov/26522470/>

Please see my response to comment 5 as well.

3) I am sure you removed SNPs, indels, etc. I am not questioning your methodology. I do not have doubts that you missed a SNP or an indel. My issue is, you cannot claim privacy based on a bioinformatics tool. For example, I get your whole genome, tempered with its format, input it to all these tools, and the tools failed. Can I claim that I provide privacy for your genome by slightly tempering its file format?

4) This comment conflicts with your previous comments. If after anonymizeBAM, only reference sequence information remains, then what remains is useless. I assume that is not what you meant here. I suggested differential privacy (DP), because DP can provide provable privacy also for the region of the genome you share (after removing SNPs). For example, if you can quantitatively show that use-case scenarios you listed above can be done in an accurate way after using DP on the shared non-SNP data, that can be a good contribution. For example, see below works about use of DP for genomic data sharing:

https://ieeexplore.ieee.org/abstract/document/6137439?casa_token=FxgA3OkorYAAAAA:2Qhf1rKRpuZLrZB-iIV05gDCXGd38dCKOXuIU2B_J0kqIY-VCfbmDSJnSBR9d-YQaTbtMq_f

https://dl.acm.org/doi/abs/10.1145/2810103.2813610?casa_token=0ITQ_vZ_BmCAAAAA:Uzoh3Io_q7CAeyBtvDJSFr4pDEv_UVqbwRka3Jxwdfi_4k1RUyBSmLi-CVXSOTIIQsTm7DLnyxfe

<https://academic.oup.com/bioinformatics/article/36/6/1696/5614817>

5) Please see my previous comments. Output of a bioinformatics tool cannot be an evidence of strong privacy. Thanks for pointing out several nice example applications that can be analyzed without needing SNP data. The thing is if non-SNP data is (even slightly) informative for such analyses, it means it can provide sensitive information about an individual. You can prove otherwise, only if you conduct a privacy analysis using known inference attacks (membership inference, attribute inference, deanonymization, etc.). To analyze your scheme against such attacks, you do not need any metadata or demographic information about the donors. Previous work (in e.g., Nature, Science, AJHG) includes examples of such privacy analyses.

6) The above mentioned work only stores data in an encrypted way. When the data is shared with a recipient, the recipient can see the non-encrypted data. Here, data includes short reads (excluding SNPs). In this sense (what the data recipient obtains at the end of the day) the outcome is exactly the same in the above mentioned paper and your work. Therefore, they are not orthogonal, they are parallel in my opinion.

Reviewer #4 (Expertise: Privacy enhancing technologies, genomic data anonymization):

The work is certainly on an important and challenging problem. However, I see several major issues in this paper as I list below:

1. First of all, unfortunately, there is no such a thing as "systematic anonymization" of genomic data (actually, we should not use that term for any type of data, since de-anonymization is always possible depending on an attacker's auxiliary knowledge).

As we point out in the manuscript, we here solve the problem of removing genetic variation that makes up most inter-individual differences, including SNPs and indels. The discussion highlights what types of genetic information are removed and what types are not removed, and similarly, what type of genetic information is currently openly shared. We still maintain that we have fully removed all data sequence variants, although as stated from the start, we do not attempt to remove the information present in the count data.

Following the discussions with the editor, we have however agreed to remove the term "anonymization" from the manuscript, to clarify that the count information may have donor-related information, although it is questionable how that could be used (or misused). In the revised manuscript, we have revised the text with more precise wording and we do not use the term "anonymization".

2. Information in the genome that identifies a donor is not something like a known region. The whole genome can be used to identify an individual depending on what the attacker knows about the individual. I am not sure which SNPs the authors are planning to mask. If they mask all SNPs, then the utility in the data will be completely lost (please see my later comment on this). If they hide some selective SNPs, there are lots of work on the literature showing the privacy risk under such selective hiding (e.g., inferring the masked SNPs from unmasked ones using the correlations between them).

The reviewer has misunderstood the purpose of this manuscript, and the need for this tool in the research community. There are many large-scale human sequencing projects where sequence data needs to be shared although the SNPs are not primarily important (for example in the Human Cell Atlas and Lifetime projects, to just name two of the largest). BAMboozle is the first tool that can efficiently and without leaving scars remove the sensitive information so that data generated from human tissues in these large consortia can be openly shared. For this use, we indeed want to remove **all** SNPs and indels, and the remaining data, often produced from functional genomics assays (e.g., RNA-seq, ATAC-seq, etc), is highly useful. The fully processed data can then be openly shared and used for the following important analysis questions (just to name a few areas of importance):

- 1) Study cell-type, sub-type and cell-state definitions and associated gene expression patterns from large-scale single-cell RNA-seq and ATAC-seq studies.
- 2) Perform meta-analysis across many data sets to improve point 1 above
- 3) Learn about alternative splicing or alternative use of promoters and polyadenylation sites for each cell, cell-type etc using population-level or single-cell RNA-sequencing.
- 4) Potential to use RNA velocity to study the directionality of cell-states by quantifying intronic and exonic reads for each gene and cell (very important for single-cell RNA-sequencing)
- 5) Potential to link RNA, ATAC-seq, splicing data for integrated analysis of the genetic programs inside each cell type etc.

6) Benchmarking studies on how different single-cell genomics methods perform in relation to other methods.

By these examples, we wish to highlight that the reviewer's claim "If they mask all SNPs, then the utility in the data will be completely lost (please see my later comment on this)" is completely misinformed.

The reviewer's comment likely arises from working with privacy enhancing technologies in areas of population genetics and the mapping genetic loci to disease. We agree that our tool is not useful in this area, which is clearly written in our manuscript (discussion). For these applications, raw data with highly protected access is required and other strategies to obfuscate data become relevant.

In the revised manuscript, we have added additional discussions to clarify these core points further to the point it should now be impossible to misunderstand the manuscript in the manner of the reviewer.

3. Privacy is validated/quantified using a bioinformatics tool only, not via established methods. There are several inference attacks on genomic data that can be used for de-anonymization. I would expect the authors at least run the obfuscated genome against such attacks to prove their statement.

We used cell-snp-lite and vireo (two widely used tools for finding donor-related genetic variation in large-scale sequence data), to demonstrate that indeed no donor-related genetic information were present in the processed data. We further used bcftools to, in another manner, demonstrate that all sequence information after BAMboozle processing is completely identical to the reference genome.

The reviewer mentions different strategies that quantifies the amount of donor-specific information in sequence data, but since we have removed all genetic sequence variants, none of the existing tools can be run.

We note however that count data may contain donor-related information, and that is an open challenge in the data privacy field to investigate whether the sharing of count data (which is currently standard in the field) has privacy concerns. However that is outside the scope of this manuscript.

4. There is the well-known differential privacy concept to obfuscate data (there are several works using this for genomes as well), which provides statistical guarantees (also against de-anonymization). But, there is no mention of this in the entire paper.

Again, the whole point of BAMboozle is remove **all** genetic variants, including SNPs and indels, rather than a subset or using a strategy to obfuscate data. The examples of differential privacy methods start by quantifying genetic sequence variants that are completely missing in our data.

5. Paper needs a detailed utility analysis. Authors claim that there are still analyses that can be done on obfuscated data. If all SNPs will be masked, I do not see much utility in the remaining data. So, I think the paper certainly requires a detailed utility analysis.

We have added significant utility analysis including the demonstration that BAMboozled data preserves:

1) cell-type specific gene expression signatures

2) splicing kinetics information, e.g. captured with the widely used RNA velocity strategy.

3) count characteristics needed for downsampling of data, often used in methods benchmarking studies.

4) CNVs

6. The works authors compare their proposed scheme with are not actually the seminal papers in this domain. I suggest the authors do a more depth literature review on this area. For example, hiding SNP information from short reads (BAM) files have been already explored. Please see the following work: https://link.springer.com/chapter/10.1007/978-3-642-54568-9_9. Also, authors should compare and position their work with existing attacks against genomic privacy (especially inference and de-anonymization attacks) and differentially-private approaches.

We thank the reviewer for mentioning the above study that we were not aware of. This citation has been added to the revised manuscript.

We note however, that the above-mentioned study focuses on an encrypted database to which queries can be made and masked sequence data is returned. This is very much in line with several ongoing efforts, although completely orthogonal to BAMboozle where the goal is to share data directly and openly after removing all SNPs and indels. With this strategy, the remaining sequence data contains no or so little genetic variation that attacks trying to connect it to a person are futile and can also not lead to additional sensitive information (eg. disease-relevant SNPs) being learned anyways.

Editorial questions

(1) Even more careful wording when reporting the anonymity of the data outputted by anonymizeBAM, as none of the analyses presented exclude the possibility of combining the remaining genomic information with other data types to infer subject identity. This point was previously brought up by R1 in some form. While the reviewer suggests you actually explore how secure the data is by combining it with other data types, we appreciate this might be difficult if you do not have the right data for that. At least the cell lines you use in the current examples might not have been reported with much other subject-specific data, in which case, it's understandable that such an analysis is -in fact- not possible.

We have followed this suggestion by removing the word anonymization from our manuscript, to clarify that although all sequence information is removed, there is information content in the count data, which is why the processed data has utility. Whether count data pose privacy concerns is an open question for the whole field of genetic privacy, despite the fact that such information is currently openly shared.

(2) An explanation of what SNPs and other genomic features were replaced, and why they were chosen.

We have clarified this core issue further by a paragraph in the results section and also an additional statement in the discussion, both pasted below:

“Importantly, the removal of SNPs and indels with BAMboozle is not dependent upon user-supplied

lists or databases of SNPs and indels, rather all single-nucleotide or indel differences relative to the reference sequence are replaced by the annotated reference bases. Thus, complete removal of SNPs and indels with BAMboozle is possible even though current databases of SNPs and indels are incomplete. This procedure effectively removes all types of genetic variant information, as sequenced reads aligning to the genome are all reverted to reference sequence, and the remaining sequenced reads that fail to align are discarded.”

(3) A utility analysis, which we believe is well within your laboratory's expertise, and is something we hope to be extended through a few examples. These examples need to demonstrate situations in which the "anonymized" data does not affect downstream analyses typically done with (sc-)RNASeq data, and one or more situation where it does (for example, exploring the clonality of patient cancers using sc-RNASeq data).

We already show that BAMboozled data can be used to identify cell type, sub-type and cell-state related gene expression profiles. Moreover, as listed in the response to the reviewer comment 2 above, the processed BAM files can be used to study alternative splicing, alternative promoter usage or polyadenylation sites usage. It can also be used to compare and benchmark various methods and analysis strategies. After removing essential genetic variation, there is however no possibility to infer clones as in cancer sequencing studies. For this purpose, the raw (and typically highly protected) data is needed.

Since we remove all SNPs and indels, the results of a utility analysis would all be binary. Many areas are not affected (e.g. those listed in response to comment 2 above), whereas other applications (e.g. associating genetic variation to disease, or somehow utilizing donor-specific genetic variation, or clonal variation within a donor) has no utility. Since these utilities are fully predictable (those relying on the donor-specific genetic variation against those that don't), a utility analysis would not have any merit.

In the revised manuscript, we have further extended the result and discussion of the utility and limitations with BAMboozled processed data.